# The transcription factor ERG regulates a low shear stress-induced anti-thrombotic pathway in the microvasculature

C. Peghaire [1], N.P. Dufton [1], M. Lang[1], I.I. Salles-Crawley [2], J. Ahnström[2], V. Kalna[1], C. Raimondi[1], C. Pericleous[1], L. Inuabasi[1], R. Kiseleva[3], V.R. Muzykantov[3], J.C. Mason[1], G.M. Birdsey [1] & A.M. Randi[1]*

Endothelial cells actively maintain an anti-thrombotic environment; loss of this protective function may lead to thrombosis and systemic coagulopathy. The transcription factor ERG is essential to maintain endothelial homeostasis. Here, we show that inducible endothelial ERG deletion ($Erg^{iEC-KO}$) in mice is associated with spontaneous thrombosis, hemorrhages and systemic coagulopathy. We find that ERG drives transcription of the anticoagulant thrombomodulin (TM), as shown by reporter assays and chromatin immunoprecipitation. TM expression is regulated by shear stress (SS) via Krüppel-like factor 2 (KLF2). In vitro, ERG regulates TM expression under low SS conditions, by facilitating KLF2 binding to the TM promoter. However, ERG is dispensable for TM expression in high SS conditions. In $Erg^{iEC-KO}$ mice, TM expression is decreased in liver and lung microvasculature exposed to low SS but not in blood vessels exposed to high SS. Our study identifies an endogenous, vascular bed-specific anticoagulant pathway in microvasculature exposed to low SS.

[1] National Heart and Lung Institute, Imperial College London, London, UK. [2] Centre for Haematology, Hammersmith Hospital Campus, Imperial College London, London, UK. [3] Department of Pharmacology, Institute for Translational Medicine and Therapeutics, University of Pennsylvania School of Medicine, Philadelphia, USA. *email: a.randi@imperial.ac.uk

Hemostasis is a physiological mechanism, which aims to maintain vessel integrity through the formation of a blood clot after injury. A crucial function of the healthy endothelium is to maintain a constitutive anti-thrombotic state via the fine regulation of many procoagulant and anticoagulant genes[1,2]. Clinical evidence supports the concept that vascular bed-specific pathways balance local hemostasis and thrombosis, with differences between arterial, venous and microvascular beds[3,4], suggesting possible advantages in vascular-bed specific therapeutic approaches. However, the molecular basis for such specificity is poorly understood.

Amongst the multiple procoagulant and anticoagulant pathways controlled by endothelial cells (EC), thrombomodulin (TM) is a key player in the regulation of coagulation and thrombosis. TM is widely expressed by EC of intact vessels and exerts anticoagulant properties by potentiating activation of protein C; it also has anti-fibrinolytic and anti-inflammatory functions[5]. In vivo, 60% of the mice with genetic constitutive deletion of TM in EC survive beyond birth, but show impairment of protein C activation and succumb to spontaneous thrombosis, resulting in consumption of coagulation factors, coagulopathy and generalised bleeding[6]. In line with its anticoagulant and anti-inflammatory roles, TM has been implicated in multiple inflammatory and ischemic pathologies. Polymorphisms in the TM gene have been reported to be associated with increased risk of venous thromboembolism[7], bleeding disorder[8], coronary artery disease[9], atypical hemolytic-uremic syndrome[10] and atherosclerotic disease[11]. TM expression was shown to be downregulated on EC overlying the atherosclerotic plaque[12]. Finally, recombinant soluble TM therapy has been shown to be beneficial in treating sepsis and suspected disseminated intravascular coagulation[13,14].

Previous studies have demonstrated that different external stimuli are able to transcriptionally regulate TM. Laminar shear stress (SS) has been shown to upregulate TM expression[15,16], whereas inflammatory cytokines such as TNF-α repress TM expression via the transcription factor (TF) NF-κB[17,18]. Multiple TF have been implicated in the regulation of TM expression, including Krüppel-like factor 2 (KLF2)[5,19]. KLF2 has been shown to regulate TM levels in EC in static, low and high SS conditions[20,21]. Upregulation of TM expression by high SS, mediated by KLF2 (refs[20–22]), is an important protective anti-thrombotic and anti-atherosclerotic mechanism in large arteries[23–25]. However, unlike loss of endothelial TM, in vivo deletion of KLF2 in mice leads to a modest decrease in TM expression and is not sufficient to cause spontaneous thrombosis, showing that KLF2 requires the cooperation with other TF to regulate TM levels and exert its anti-thrombotic effects[26], possibly in selected vascular beds.

The TF ETS-related gene (ERG) is the most abundant member of the E-26 transformation specific (ETS) family[27] in adult differentiated EC, expressed in all vascular beds[28,29]. Multiple studies have described the essential role of ERG in the endothelium (reviewed previously[27]). ERG regulates vascular development and angiogenesis through pathways including Notch and Wnt signalling. In the adult, ERG promotes vascular homeostasis through multiple processes[27,30–33]. ERG represses vascular inflammation in healthy endothelium, through inhibition of NF-κB p65 binding to the promoters of pro-inflammatory genes[34,35]. Moreover, ERG protects the endothelium from endothelial-to-mesenchymal transition (EndMT), a process associated with chronic inflammation and tissue damage in diseases such as end-stage liver fibrosis[36].

Given its multiple homeostatic functions, we hypothesise that ERG may have an anti-thrombotic effect in EC, thus protecting blood vessels from spontaneous clot formation. In this study, we show that inducible chronic endothelial-specific deletion of ERG in adult mice leads to systemic coagulopathy with spontaneous thrombosis and/or hemorrhages in selected vascular beds. We find that ERG regulates TM expression in vivo and that treatment with recombinant TM fusion protein partly rescues the thrombotic phenotype observed in ERG-deficient mice. In vitro, we demonstrate that ERG binds to and transactivates the TM promoter. We identify an ERG-KLF2 complex which cooperates in driving TM expression. ERG facilitates KLF2 chromatin access, DNA binding and transactivation of the TM promoter. Interestingly, this pathway is selective for low SS conditions both in vitro and in vivo. These results identify a low SS-dependent transcriptional pathway providing protection from spontaneous thrombosis in the microvasculature.

## Results

**ERG deletion in mice results in thrombosis and coagulopathy.** Previous studies have shown that deletion of endothelial ERG in adult mice causes defective angiogenesis, loss of endothelial homeostasis and enhanced tissue inflammation in multiple organs[27,37]. To investigate whether ERG protects EC from thrombosis in vivo, we used adult mice with inducible endothelial deletion of ERG (Erg[iEC-KO])[30,33]. Erg[iEC-KO] mice were injected with tamoxifen at 6–8 weeks old and analysed 30 and 45 days after injection. Effective downregulation of ERG expression post-tamoxifen injection was confirmed by qPCR on liver and lung lysates (Supplementary Fig. 1a, b). Histological analysis of multiple tissues revealed the presence of hemorrhages in liver and lung in Erg[iEC-KO] mice, as shown by H&E staining (Fig. 1a, b) and Masson's Trichrome (Supplementary Fig. 1c). Histological analysis also showed the presence of spontaneous clots in the liver (Fig. 1c), in 78% (7/9) and 100% (4/4) of Erg[iEC-KO] mice, at 30 and 45 days post-tamoxifen injection, respectively. No clots were identified in lung, brain or kidney tissues. Immunofluorescence (IF) microscopy confirmed spontaneous intravascular thrombi in liver blood vessels, with high fibrinogen/fibrin content (Fig. 1d) and platelet aggregates (Fig. 1e), demonstrating that endothelial ERG protects from thrombosis in the liver.

To determine whether endothelial ERG impacts the coagulation cascade, a thrombin generation assay was performed on mouse plasma from Erg[iEC-KO] mice and littermate control mice (Supplementary Fig. 1d–f). The data showed no significant change in the endogenous thrombin potential (ETP) (Supplementary Fig. 1e), but a significant increase in thrombin maximum peak height (Supplementary Fig. 1f). Interestingly, mice generating the highest concentration of thrombin in plasma showed the most pronounced thrombotic phenotype in the liver.

Flow cytometry performed 30 and 45 days post-tamoxifen (Fig. 1f and Supplementary Fig. 1g, respectively), revealed a significant decrease in platelet counts in Erg[iEC-KO] mice; expression of the platelet marker GP1bβ was unchanged (Supplementary Fig. 1h). Crucially, analysis of mouse plasma 30 days post tamoxifen showed decreased fibrinogen concentration (Fig. 1g) associated with elevated D-dimer (Fig. 1h) and thrombin-antithrombin (TAT) complex (Fig. 1i) levels in Erg[iEC-KO] mice; these findings are indicative of coagulopathy. Notably, the penetrance and severity of the phenotype in Erg[iEC-KO] mice appeared to be heterogeneous and variable (Supplementary Fig. 2a, b).

These data show that the endothelial TF ERG is essential to protect from spontaneous thrombosis in selected vascular beds. Deletion of ERG disrupts regulation of coagulation, leading to a coagulopathy associated with thrombosis and/or bleeding in liver and lung.

**ERG controls expression of multiple regulators of coagulation.** To investigate the molecular mechanisms through which ERG exerts its anti-thrombotic function, we analysed the global transcriptome profile of ERG-deficient HUVEC[38] and identified

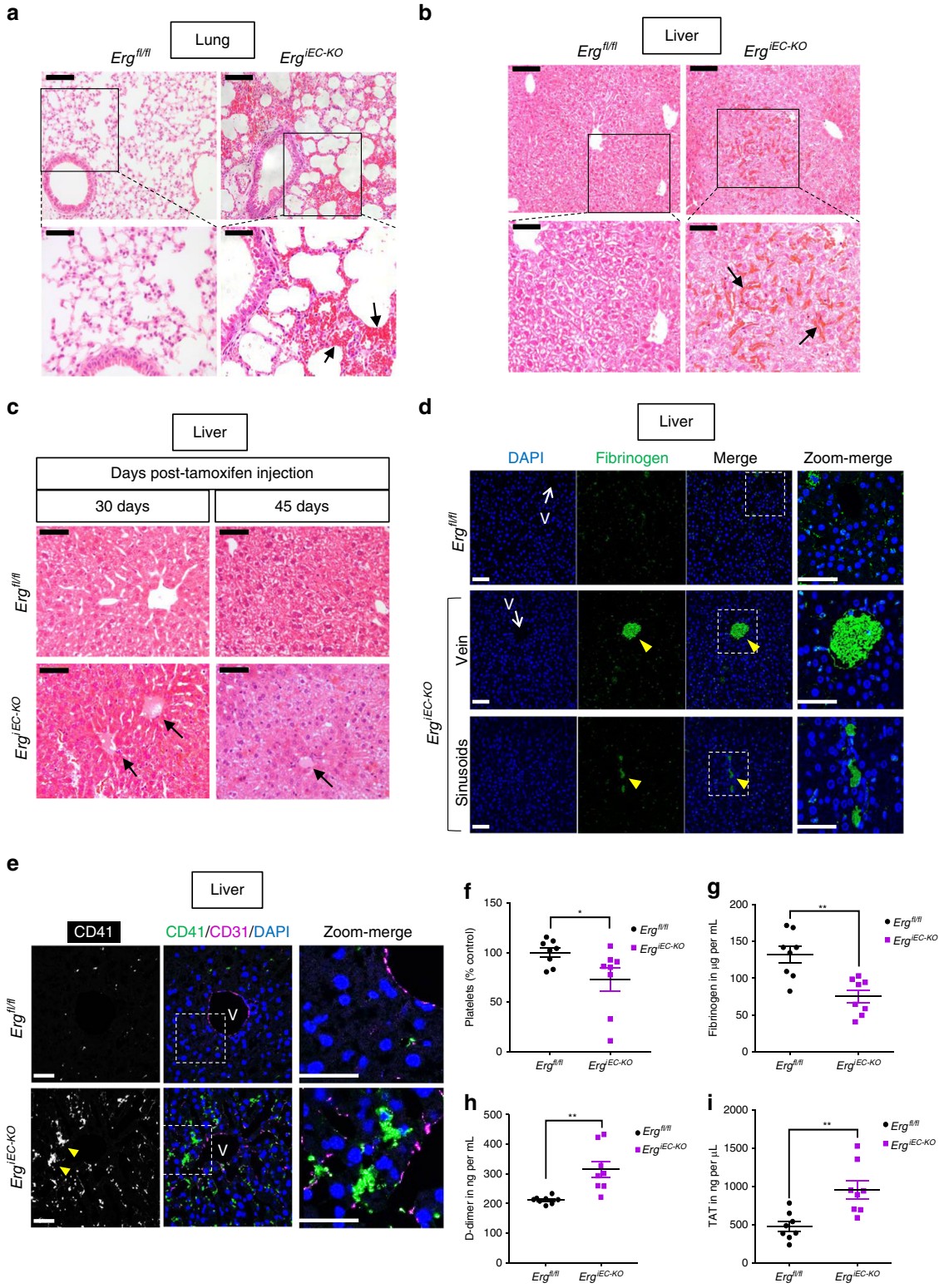

multiple genes directly or indirectly involved in the regulation of coagulation as putative ERG transcriptional targets. These genes were validated by qPCR in HUVEC (Fig. 2a). Interestingly, in vivo analysis showed a tissue-specific pattern in the ERG-dependent regulation of hemostatic genes (including *FVIII*, *PAI1* and *eNOS*) by ERG (Fig. 2b, c). The most consistently regulated ERG target gene, both in vivo and in vitro, is *TM*. *TM* mRNA was significantly downregulated in both liver and lung lysates from

ERG-deficient mice (Fig. 2d, e). This was not due to the loss of vasculature, since expression of endothelial markers (*CD31*, *VWF*) in lung and liver tissues was not significantly different in *Erg*^*iEC-KO*^ mice compared to control (Fig. 2b, c). Immunoblotting of lung lysates from *Erg*^*iEC-KO*^ mice confirmed that TM protein was significantly downregulated compared to control mice (Fig. 2f). These findings suggest that ERG may regulate multiple procoagulant and anticoagulant genes in a tissue-specific manner

**Fig. 1** Endothelial-specific deletion of ERG in adult mice results in thrombosis and coagulopathy. **a, b** Representative images of H&E staining in **a** liver and **b** lung sections from $Erg^{iEC-KO}$ and $Erg^{fl/fl}$ mice showing the presence of extravasated red blood cells (black arrows) in $Erg^{iEC-KO}$ mice, 30 days after tamoxifen injection. Scale bar: 100 μm for top panels and 50 μm for bottom panels. **c** Liver sections from $Erg^{iEC-KO}$ and $Erg^{fl/fl}$ mice imaged following H&E staining, 30 and 45 days after tamoxifen injection, revealed the presence of spontaneous thrombi in liver from $Erg^{iEC-KO}$ mice, as shown by clots (black arrows) partially occluding veins. Scale bar 50 μm. **d, e** Liver sections from $Erg^{iEC-KO}$ mice, 30 days after tamoxifen injection, stained with **d** an anti-fibrin/fibrinogen (green) antibody and **e** a platelet marker (CD41 antibody, green) confirmed the presence of clots (yellow arrowheads) with high fibrinogen content and platelet aggregates in veins (V) and in sinusoids. Tissues are co-stained for CD31 to visualise blood vessels (magenta); nuclei are identified by DAPI (blue). Scale bar 50 μm. **f** Platelet counts (Platelets x $10^3$ per μl expressed as % of control mice) were determined on plasma from $Erg^{iEC-KO}$ and $Erg^{fl/fl}$ mice, 30 days post tamoxifen ($n = 8$ per genotype). **g** Fibrinogen concentration (μg per mL) was measured on plasma from $Erg^{iEC-KO}$ and $Erg^{fl/fl}$ mice, 30 days post tamoxifen, using mouse Fibrinogen ELISA kit ($n = 8$ per genotype). **h** D-dimer concentration (ng per mL) was measured on plasma from $Erg^{iEC-KO}$ and $Erg^{fl/fl}$ mice, 30 days post tamoxifen, using mouse D-dimer ELISA kit ($n = 8$ per genotype). **i** Thrombin-antithrombin (TAT) complex concentration (ng per μL) was measured on plasma from $Erg^{iEC-KO}$ and $Erg^{fl/fl}$ mice, 30 days post tamoxifen, using mouse TAT ELISA kit ($n = 8$ per genotype). All graphical data are mean ± s.e.m., *$P < 0.05$, **$P < 0.01$, ***$P < 0.001$, Student's $t$-test. Source data are provided as a Source Data file

and points to TM as a key candidate for the anti-thrombotic effects of ERG.

**Thrombomodulin treatment rescues $Erg^{iEC-KO}$ mice phenotype**. To test whether the thrombotic phenotype observed in $Erg^{iEC-KO}$ mice was due to the decrease in TM expression, we used a mouse recombinant TM fusion protein targeting red blood cells (RBC-TM)[39] to restore TM protein activity in vivo. RBC-TM has been shown to have a longer half-life and to confer more durable and potent protection from thrombosis in vivo compared to soluble non-targeted thrombomodulin (sTM)[39]. The ability of RBC-TM to generate activated protein C (APC) was confirmed in vitro (Supplementary Fig. 3a). Plasma was collected from adult control and $Erg^{iEC-KO}$ mice (25 days post-tamoxifen injection) before and after (6 h) injection of RBC-TM (4 mg per kg, intravenous route). Confirmation of ERG deletion and decrease in TM levels were assessed on liver (Fig. 3a and Supplementary Fig. 3b, d) and lung (Fig. 3b and Supplementary Fig. 3c, e) tissues. Crucially, acute treatment with RBC-TM in $Erg^{iEC-KO}$ mice was able to rescue TAT levels (Fig. 3c), an early systemic plasma marker for coagulopathy/thrombosis with short half-life (45 mins). As expected, treatment with RBC-TM for 6 h was not sufficient to correct levels of D-dimer (Fig. 3d) and fibrinogen (Fig. 3e), which are circulating biomarkers of coagulopathy that emerge later and have longer half-lives. These data show that an acute treatment with RBC-TM is able to significantly rescue the coagulopathy observed in $Erg^{iEC-KO}$ mice.

**ERG regulates thrombomodulin expression in endothelial cells**. We used in vitro models to investigate the mechanisms through which ERG controls TM expression. Inhibition of ERG expression in HUVEC by siRNA treatment, for 12, 24 and 48 h, caused a decrease in TM mRNA (Fig. 4a and Supplementary Fig. 4a) and protein expression, assessed by immunoblotting (Fig. 4b, c and Supplementary Fig. 4b) and IF microscopy (Fig. 4d, e), compared to control siRNA. Consistently, ERG inhibition via a second siRNA sequence confirmed downregulation of $TM$ mRNA expression levels (Supplementary Fig. 4c). ERG also regulates TM expression at the mRNA and protein levels in microvascular EC, in both human dermal microvascular EC (HDMEC) (Supplementary Fig. 4d and Fig. 4d) and human dermal blood EC (HDBEC) (Fig. 4e). Conversely, ERG overexpression (Supplementary Fig. 4e, f) significantly increased $TM$ expression (Fig. 4f) in HUVEC.

TM is a key regulator of thrombin-mediated activation of the natural anticoagulant APC[5]. To investigate whether the regulation of TM expression by ERG results in regulation of its function, we conducted a protein C activation assay on HUVEC treated with control or ERG siRNA and measured the production

of APC over time. In line with decreased TM expression, ERG-deficient cells (Supplementary Fig. 4g, h) showed reduced generation of APC after 30 min of incubation with protein C, with a maximum reduction at the 60 min time point compared to siCtrl cells (Fig. 4g).

These results demonstrate that ERG regulates TM levels and thus the TM-protein C system in EC.

**ERG transcriptionally regulates thrombomodulin expression**. We investigated the molecular mechanisms by which ERG induces TM expression in EC. Bioinformatic analysis of the human $TM$ gene locus revealed the presence of highly conserved ERG DNA binding motifs in the promoter region (Fig. 5a and Supplementary Fig. 5a–c). Comparative analysis of chromatin immunoprecipitation sequencing (ChIP-seq) data for markers of active promoters (Histone marks H3K4Me3/H3K27Ac and RNA pol II) in HUVEC (available from ENCODE Consortium)[40] showed that the location of these marks correlates with the position of the putative ERG binding sites (Fig. 5a). Data from ERG ChIP-sequencing[41] (Fig. 5a and Supplementary Fig. 5a; ERG binding peak) and ChIP-qPCR (Fig. 5b, TM promoter regions R1 and R2; sequence of ERG binding peak shown in Supplementary Fig. 5c) confirmed ERG binding to the TM promoter but not in a negative control region distal to the ERG binding peak (Fig. 5b, TM negative control). Specificity of the ERG antibody was confirmed in ERG-deficient HUVEC (Fig. 5b). Functionality of the ERG binding sites (EBS; A/CGGAA) (Supplementary Fig. 5b) present on the proximal TM promoter was shown by luciferase reporter assays (sequence of the TM promoter construct shown in Supplementary Fig. 6). ERG overexpression resulted in transactivation of TM wild-type (TM WT) promoter in HUVEC (Fig. 5c); mutation of the two EBS located upstream the transcription start site (TSS) of the $TM$ gene were sufficient to abrogate this signal, showing that these EBS are functional and are responsible for ERG's ability to drive TM (Fig. 5c and Supplementary Fig. 6).

The acetyltransferase p300 has been shown to bind to the TM promoter and activate its expression[18]. Pharmacological inhibition of p300 in HUVEC, resulting in decreased global levels of histone 3 lysine (K) 27 acetylation (H3K27Ac) (Supplementary Fig. 7a), led to a decrease in H3K27Ac occupancy on the TM promoter (Fig. 5d) and a decrease in $TM$ mRNA levels (Supplementary Fig. 7b). We have recently shown that ERG is involved in p300 recruitment and acetylation of H3K27 on super-enhancers associated with endothelial cell-specific genes[41]. In line with this concept, we showed that ERG deficient-HUVEC have reduced p300 binding to the TM promoter region (Fig. 5e, regions R1 and R2). Deletion of ERG also led to a significant decrease in H3K27Ac occupancy, in agreement with a reduction in open and active chromatin (Fig. 5f, regions R1 and R2),

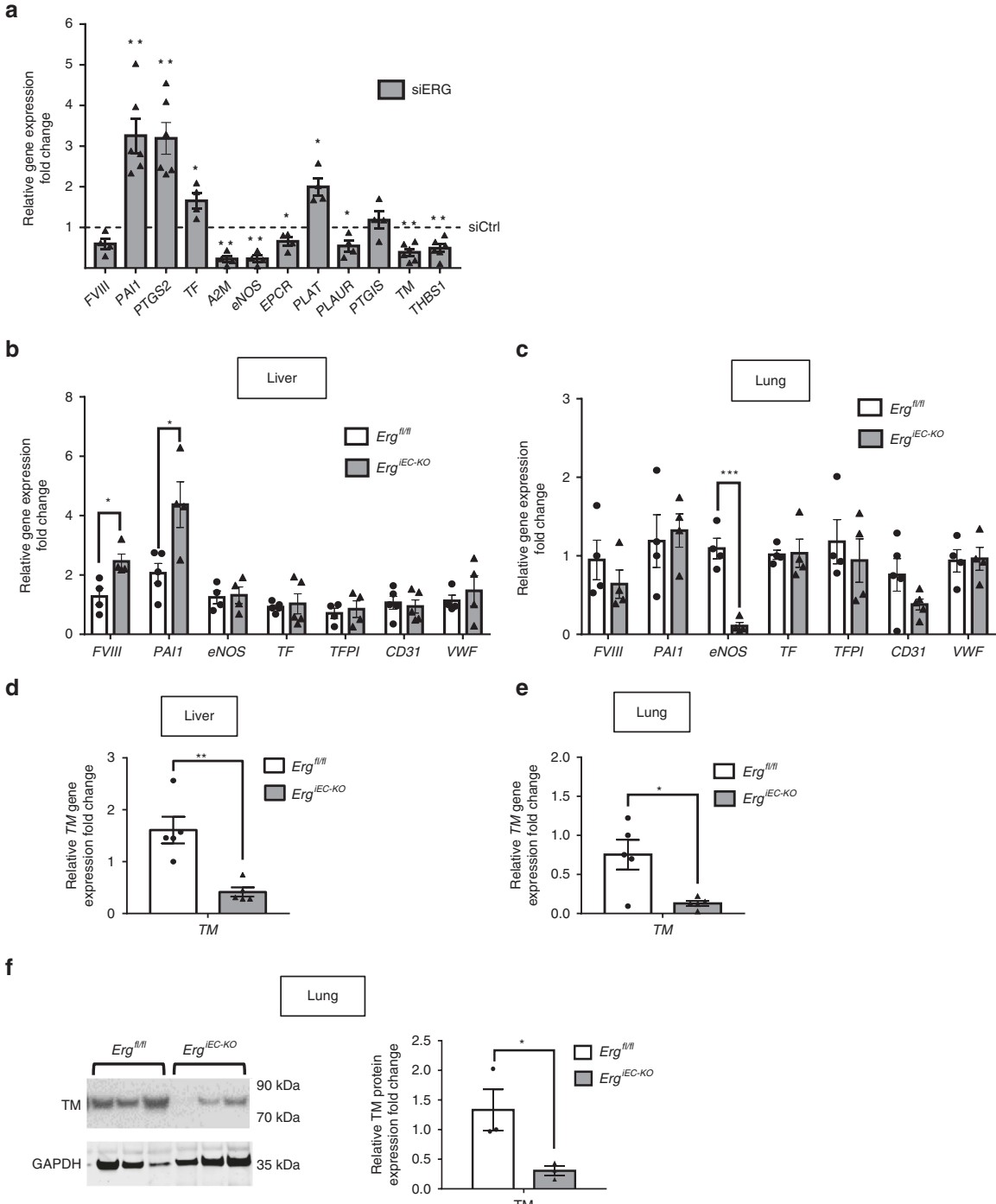

**Fig. 2** ERG controls the expression of multiple regulators of coagulation in a tissue-specific manner. **a** qPCR screen analysis of genes involved in the regulation of coagulation in HUVEC was performed at 48 h after ERG inhibition by siRNA ($n = 4$–6 independent experiments). Data were normalised to *GAPDH*. **b**, **c** qPCR analysis of selected targets genes directly or indirectly involved in coagulation and endothelial markers in whole liver (**b**) and lung (**c**) lysates from adult control (*Erg*<sup>fl/fl</sup>) and ERG-deficient (*Erg*<sup>iEC-KO</sup>) mice ($n = 4$–5 per genotype). Data were normalised to *18S*. **d**–**e** qPCR analysis of thrombomodulin (*TM*) mRNA expression in whole liver (**d**) and lung (**e**) lysates from adult *Erg*<sup>fl/fl</sup> and *Erg*<sup>iEC-KO</sup> mice ($n = 5$ per genotype). Data were normalised to 18S. **f** Immunoblot and quantification of WB for TM in protein extracts of lung samples from adult *Erg*<sup>fl/fl</sup> and *Erg*<sup>iEC-KO</sup> mice 30 days after tamoxifen injection ($n = 3$ mice per group). Data were normalized to GAPDH. All graphical data are mean ± s.e.m, *$P < 0.05$, Student's $t$-test. Source data are provided as a Source Data file

suggesting that ERG directly affects chromatin accessibility at the TM promoter. These data demonstrate that ERG binds to the TM promoter and directly drives *TM* expression in EC, through p300 recruitment, acetylation of H3K27 and opening of the chromatin at the TM promoter region.

**ERG cooperates with KLF2 to drive thrombomodulin expression.** The TF KLF2 has been identified as a key regulator of TM expression in EC[20]. Analysis of the ERG bound sequence on the TM promoter revealed the presence of adjacent ERG and KLF2 binding sites close to the TSS (Supplementary Fig. 5c). Thus, we

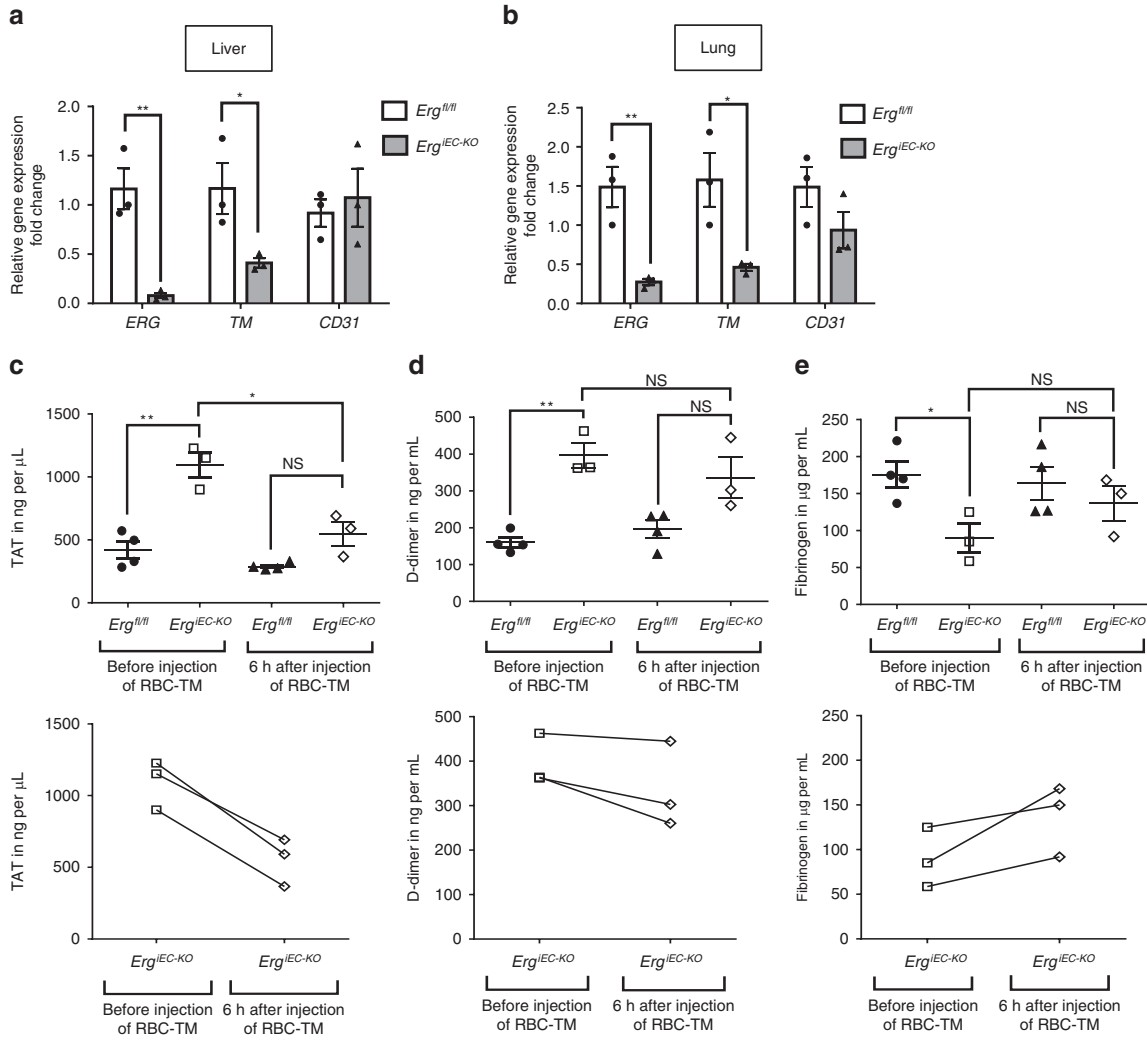

**Fig. 3** Treatment with RBC-thrombomodulin rescues the thrombotic phenotype of $Erg^{iEC-KO}$ mice. **a, b** qPCR analysis of *ERG*, *CD31* and *TM* gene expression in whole **a** liver and **b** lung lysates from adult control ($Erg^{fl/fl}$) and ERG-deficient ($Erg^{iEC-KO}$) mice ($n = 3$); 25 days after tamoxifen injection and 6 h after RBC-TM injection. Data were normalized to 18S. Graphical data are mean ± s.e.m., *$P < 0.05$, Student's *t*-test. **c–e** Adult $Erg^{iEC-KO}$ and control mice were bled before and after (6 h) injection of red blood cells-targeted thrombomodulin (RBC-TM, 4 mg per kg) to analyse plasma markers for coagulopathy. **a** Thrombin-antithrombin (TAT) complex concentration (ng per μL) was measured on plasma from $Erg^{iEC-KO}$ and $Erg^{fl/fl}$ mice, using mouse TAT ELISA kit ($n = 3$–4 per genotype). **b** D-dimer concentration (ng per mL) was measured on plasma from $Erg^{iEC-KO}$ and $Erg^{fl/fl}$ mice, using mouse D-dimer ELISA kit ($n = 3$–4 per genotype). **c** Fibrinogen concentration (μg per mL) was measured on plasma from $Erg^{iEC-KO}$ and $Erg^{fl/fl}$ mice, using mouse Fibrinogen ELISA kit ($n = 3$–4 per genotype). All graphical data are mean ± s.e.m., NS: Not significant, *$P < 0.05$, **$P < 0.01$, Student's *t*-test. Data for each $Erg^{iEC-KO}$ mouse before and after RBC-TM injection are also presented in separated graphs (bottom graphs) to see individual response to the treatment. Source data are provided as a Source Data file

investigated whether ERG and KLF2 cooperate in EC to promote TM expression.

Single ERG or KLF2 siRNA treatment or a combination of the two led to a comparable decrease in TM mRNA expression (Supplementary Fig. 8a) or protein levels (Supplementary Fig. 8b, c), showing no additive effect. ERG siRNA did not affect KLF2 mRNA (Supplementary Fig. 8d) and protein levels (Supplementary Fig. 8e); similarly, KLF2 siRNA did not affect ERG levels (Supplementary Fig. 8a–c). However, inhibition of KLF2 expression caused a significant decrease in ERG's ability to drive TM promoter activity (Fig. 6a) and to promote *TM* expression (Supplementary Fig. 8f). Moreover, inhibition of ERG expression caused a significant decrease in the KLF2-dependent transactivation of the TM promoter and of *TM* expression (Fig. 6b and Supplementary Fig. 8g, h). Finally, co-expression of ERG and KLF2 significantly increased transactivation of the TM promoter compared to ERG or KLF2 alone (Fig. 6c). These findings indicate

that ERG and KLF2 co-operate in driving TM expression. Interestingly, mutations of the two ERG binding sites (EBS) upstream of the TSS, which abolished ERG-dependent transactivation (see Fig. 5c), also led to a complete loss of KLF2 transactivation ability (Fig. 6d) suggesting that ERG binding to DNA is required for KLF2 transcriptional activity. ChIP-qPCR analysis showed binding of ERG and KLF2 to the same promoter regions (Fig. 6e, f, regions R1 and R2,). Inhibition of KLF2 expression in HUVEC had no effect on ERG enrichment on the TM promoter region (Fig. 6e, regions R1 and R2). However, ERG deletion led to a significant reduction in KLF2 occupancy on the TM promoter (Fig. 6f, regions R1 and R2), showing that ERG is required for KLF2 interaction with the TM promoter.

These data suggested that the two proteins might interact. We tested this hypothesis in HUVEC, using proximity ligation assay (PLA); this revealed that ERG and KLF2 bind and form a complex located in the nucleus (Fig. 6g and Supplementary

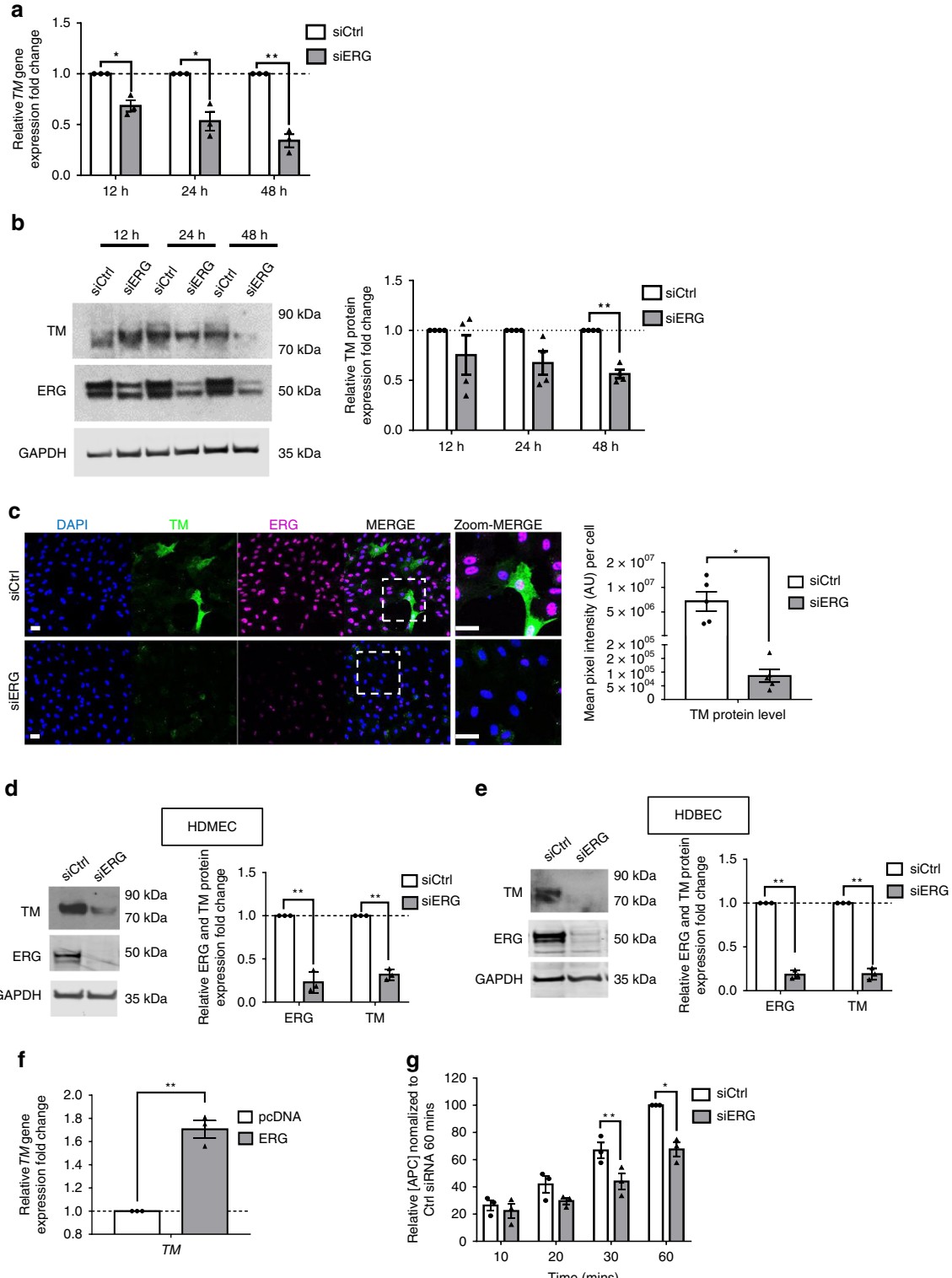

Fig. 9a–c). These data suggest a model of cooperativity where, in homeostatic conditions, ERG promotes opening of the chromatin at the TM promoter via p300 recruitment and supports KLF2 recruitment to DNA via direct interaction (Fig. 6h).

**ERG regulates thrombomodulin in low shear stress conditions.**
Although expressed in EC under static conditions in vitro, *KLF2* and *TM* are important flow-responsive genes: laminar

shear stress (SS) upregulates expression of KLF2, which then promotes increased TM expression[15,21]. Whether ERG is a flow-responsive TF is unknown. We first interrogated the effect of low (5 dynes per $cm^2$) and high (20 dynes per $cm^2$) laminar SS on ERG and KLF2 protein levels in HUVEC. As expected, KLF2 protein levels were upregulated in HUVEC exposed to low and high SS compared to static conditions; in contrast, ERG protein levels were unaffected by SS (Fig. 7a). ERG-KLF2 complex formation was increased under low SS compared to

**Fig. 4** ERG regulates thrombomodulin expression and activity in vitro. **a** qPCR analysis of thrombomodulin (*TM*) mRNA expression in control (siCtrl) and ERG-deficient (siERG) HUVEC after 12, 24 and 48 h treatment (*n* = 3 independent experiments). Data were normalized to *GAPDH*. **b** Representative immunoblot and quantification of TM following siCtrl or siERG treatment on HUVEC for 12, 24 and 48 h (*n* = 4 independent experiments). Data were normalized to GAPDH. **c** Representative image and quantification of TM expression (green) in HUVEC transfected with siCtrl or siERG siRNA for 48 h by immunofluorescence; nuclei are identified by DAPI (blue) and cells are co- stained for ERG (magenta). Scale bar 40 μm. Quantification represents the mean pixel intensity for TM signal (arbitrary unit, A.U.) per cell. **d**, **e** Representative immunoblot and quantification of ERG and TM expression in control (siCtrl) and ERG-deficient (siERG) **d** HDMEC (microvascular EC) or **e** HDBEC (microvascular EC) after 48 h siRNA treatment (*n* = 3). **f** qPCR analysis of *TM* mRNA expression in HUVEC transfected with control pcDNA or ERG cDNA expression plasmid (ERG) (*n* = 3). Data were normalised to *GAPDH*. **g** siCtrl or siERG-treated HUVEC for 48 h were incubated with protein C (300 nM), CaCl$_2$ (5 mM), thrombin (10 nM). After 10, 20, 30 and 60 min of incubation at 37 °C, anti-thrombin III (100 nM) and heparin (15 U per ml) were added to neutralise thrombin, and protein C activity was measured using chromogenic substrate S-2366 (0.5 mM) (*n* = 3). Data are expressed as relative activated protein C (APC) concentration normalised to siCtrl-treated condition following 60 min of incubation. All graphical data are mean ± s.e.m., *$P$ < 0.05, **$P$ < 0.01, ***$P$ < 0.001, Student's *t*-test. Source data are provided as a Source Data file

static conditions; high SS did not further increase this interaction (Fig. 7b).

We next investigated the role of ERG in the regulation of TM expression in static *versus* low and high SS conditions in HUVEC treated with either ERG or control siRNA. As expected, *KLF2* and *TM* mRNA levels were upregulated in HUVEC exposed to SS compared to static conditions, whilst *ERG* mRNA levels were unaffected (Fig. 7c). Crucially, whilst high SS induced upregulation of *TM* expression in ERG-deficient cells, low SS was not able to induce *TM* upregulation in ERG-deficient cells (Fig. 7c), indicating that ERG controls *TM* expression selectively in conditions of low SS and is dispensable for the high SS-dependent regulation of TM. These results show that the cooperation between ERG and KLF2 is essential for TM expression specifically in low SS conditions and suggest that other partners/pathways support KLF2-dependent transcription in high SS conditions.

HUVEC are the most widely used EC model in vitro but do not recapitulate all the properties of EC from different vascular beds. Therefore, to investigate the role of ERG in the regulation of TM in EC from microvascular or arterial origins, we conducted flow experiments on human microvascular EC (HDBEC) (Fig. 7d) and human aortic endothelial cells (HAEC) (Supplementary Fig. 10). The experiments on HDBEC showed similar results compared to HUVEC, demonstrating that ERG regulates *TM* expression specifically in low SS conditions also in microvascular EC (Fig. 7d). However, loss of ERG in HAEC did not affect low and high SS-induced upregulation of *TM* expression (Supplementary Fig. 10), indicating that ERG is not involved in the regulation of *TM* expression in these cells. These data indicate the presence of distinct molecular mechanisms of gene regulation in EC from different vascular beds. Interestingly, these differences are retained ex vivo, possibly via epigenetic memory of the cells being exposed to high SS in vivo.

High laminar SS has been shown to mediate chromatin remodelling by inducing acetylation of histone marks, thus promoting flow-dependent regulation of gene expression[42]. We have recently shown that ERG is required for acetylation of H3K27 in HUVEC in static conditions[41]; this was confirmed by decreased levels of H3K27Ac, assessed by IF microscopy, in ERG-deficient HUVEC (Fig. 7e). Here, we asked whether ERG promotes chromatin remodelling and accessibility under different SS conditions using H3K27Ac global levels as a readout. As expected, high SS increased levels of H3K27Ac compared to low SS (Fig. 7e). Interestingly, H3K27Ac levels were significantly decreased in ERG-siRNA treated cells exposed to low SS compared to control HUVEC, but were not affected by the loss of ERG in high SS conditions (Fig. 7e). These data show that ERG is required for chromatin accessibility in low SS conditions but is not required under high SS.

**ERG controls thrombomodulin expression in microvascular beds.** Finally, we investigated whether ERG regulates TM expression in EC in vivo, using adult *Erg*[iEC-KO] and control mice. IF microscopy analysis of liver and lung tissues confirmed that ERG was efficiently deleted in all vessels, including capillaries, arteries and veins from *Erg*[iEC-KO] mice (Supplementary Fig. 11a, b). CD31 (Fig. 8a, b), Isolectin B4 (Supplementary Fig. 11a, b) and wheat germ agglutinin (WGA) (Supplementary Fig. 11c, d) staining showed that despite some disruption of the vasculature in both liver and lung, deletion of ERG did not lead to significant loss of vascular coverage. TM protein levels were significantly reduced in liver sinusoids (Fig. 8a, b) and lung capillaries (Fig. 8c, d) from *Erg*[iEC-KO] mice.

Crucially, whilst ERG expression was also efficiently deleted in EC lining the descending aorta (Supplementary Fig. 12a) of *Erg*[iEC-KO] mice, TM expression was unchanged (Fig. 8e, f and Supplementary Fig. 12a, b). These findings suggest that ERG regulates TM expression specifically in microvascular beds exposed to lower SS[43,44] but not in EC of blood vessels (aorta) exposed to high SS[45,46].

In summary, this study identifies a vascular bed-specific anti-thrombotic pathway, effective selectively in low SS conditions and regulated by an ERG-KLF2 transcriptional complex. In the lung and liver microvasculature exposed to lower SS, ERG is required to provide access to chromatin and for KLF2-dependent regulation of TM expression. In larger blood vessels such as the aorta, high SS mediates epigenetic modifications, increasing access to chromatin[47]; in this context, ERG becomes dispensable for the regulation of TM. This double mechanism may provide different levels of anti-thrombotic protection: a baseline, plastic control in the microvasculature, where clot formation may be required physiologically for vascular integrity, and a higher level of control in large arteries, where the formation of a single clot can be fatal. A diagram summarising this model is shown in Fig. 9.

## Discussion

In the present study, we identify a pathway that controls the anti-thrombotic properties of the endothelium, via the transcription factor ERG.

Previous studies have shown that ERG is required for maintenance of the endothelial lineage and to promote vascular homeostasis, by controlling multiple functions including survival, permeability and inflammation (reviewed previously[27]). Here, we show that ERG also drives pathways that prevent spontaneous clotting and thus exerts anti-thrombotic properties in EC. Other TF have been reported to control pro- and anti-thrombotic pathways, and their deletion or overexpression has been shown to affect thrombus formation in a variety of in vivo injury models[26]. However, the finding that loss of a single endothelial TF is

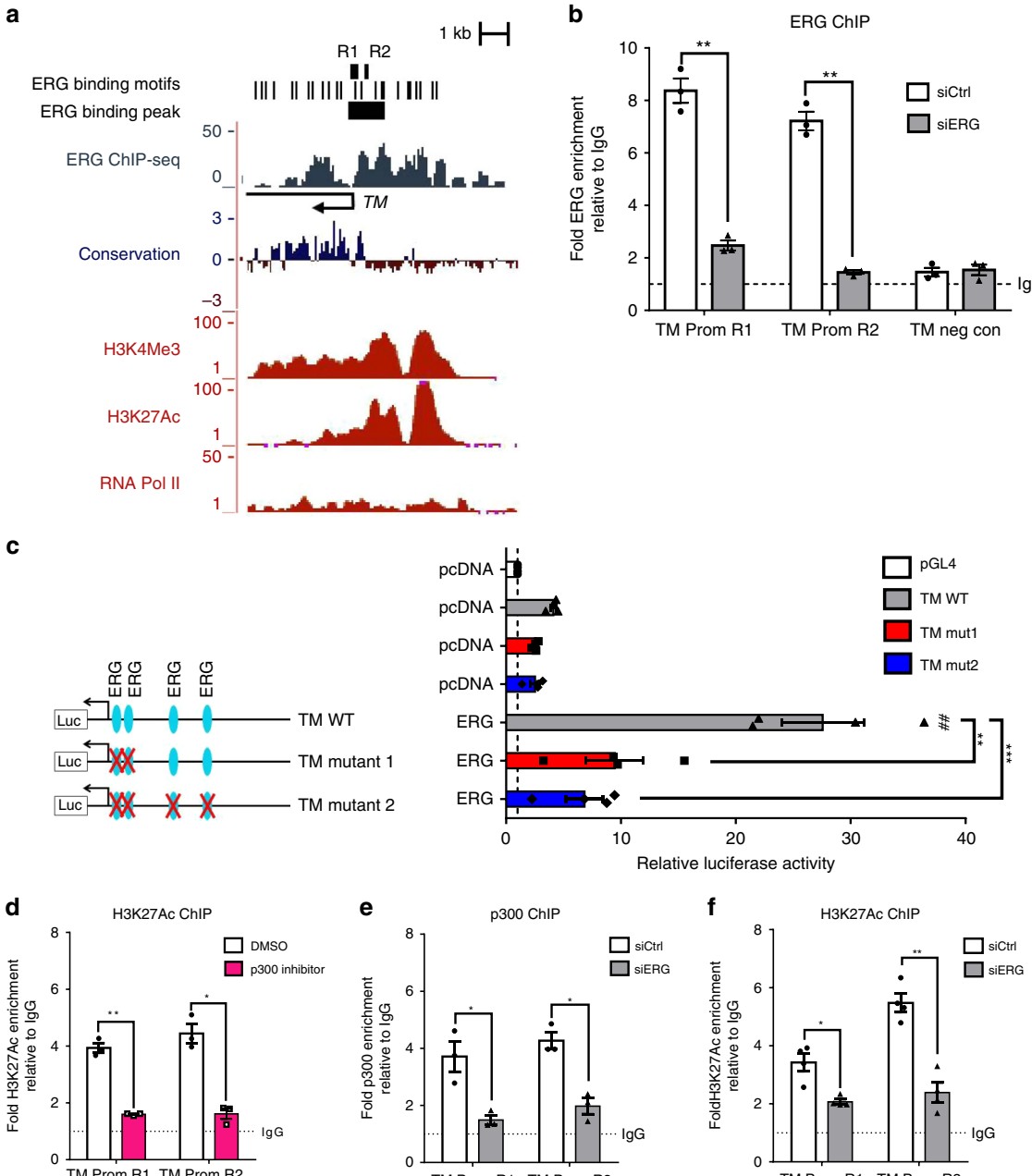

**Fig. 5** ERG binds to and transactivates the thrombomodulin promoter in vitro. **a** Putative ERG binding sites (black bars) are located within the TM promoter upstream and downstream of the transcription start site (black arrow); ERG ChIP-sequencing data show a significant ERG peak on TM proximal promoter. ENCODE sequence conservation between 100 vertebrates is shown across this gene. ENCODE ChIP-seq data profiles for H3K4Me3 (tri-methylation of lysine (K) 4 on histone 3), H3K27Ac (acetylation of lysine (K) 27 on histone 3) and RNA polymerase II (RNA pol II) in HUVEC indicate open chromatin and active transcription. Location of qPCR primers covering regions R1 and R2 (black bar) are indicated. **b** ChIP-qPCR using primers to region R1 and R2 on ERG-bound chromatin from siCtrl or siERG-treated HUVEC. Primers for a region within 5′UTR region of TM gene (TM neg con) were used as negative control. Data are shown as fold change over IgG ($n = 3$ independent experiments). Graphical data are mean ± s.e.m., *$P < 0.05$, **$P < 0.01$, ***$P < 0.001$, Student's $t$-test. **c** TM promoter luciferase reporter assay. ERG cDNA expression plasmid (ERG) or empty expression plasmid (pcDNA) were co-transfected with TM promoter-luciferase constructs (TM wild type (WT), TM mutant 1 or TM mutant 2) or a pGL4 empty vector in HUVEC, and luciferase activity was measured. Values represent the fold change in relative luciferase activity over the empty pGL4 vector alone ($n = 4$ independent experiments). Graphical data are mean ± s.e.m., **$P < 0.01$, ***$P < 0.001$, One-way ANOVA; ##$P < 0.01$ compared to pcDNA-TM WT condition, Student's $t$-test. **d** ChIP-qPCR using primers to region R1 and R2 on H3K27Ac-bound chromatin from HUVEC treated with DMSO or p300 inhibitor (10 μM) for 1 h. Data are shown as fold change over IgG ($n = 3$ independent experiments). **e-f** ChIP-qPCR using primers to region R1 and R2 on **e** p300- ($n = 3$ independent experiments) or **f** H3K27Ac-bound ($n = 4$ independent experiments) chromatin from siCtrl or siERG-treated HUVEC. Data are shown as fold change over IgG. All graphical data for ChIP experiments are mean ± s.e.m., *$P < 0.05$, **$P < 0.01$, ***$P < 0.001$, Student's $t$-test. Source data are provided as a Source Data file

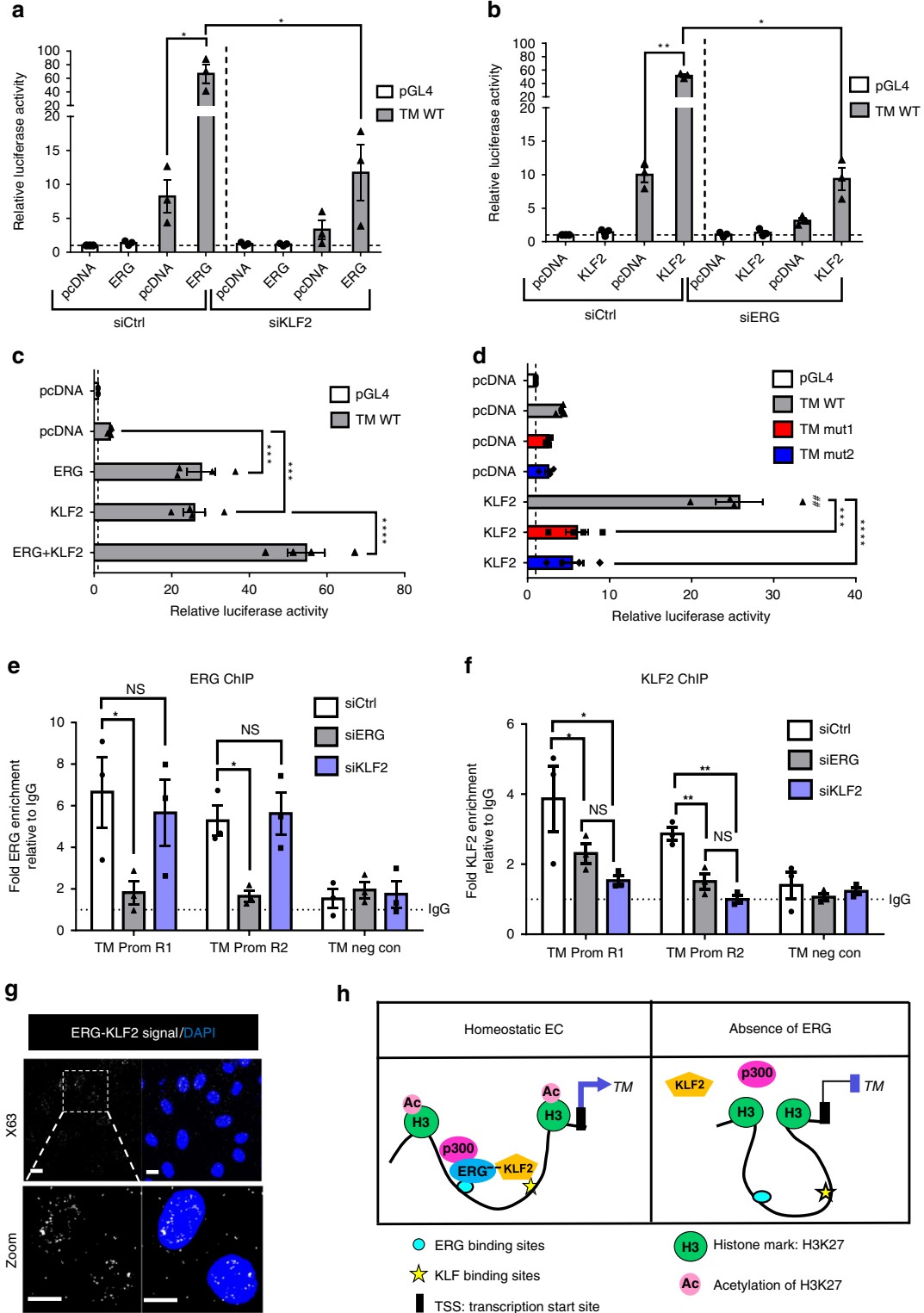

sufficient to cause spontaneous thrombosis is to our knowledge unique, and in line with the essential role of ERG in maintaining vascular homeostasis.

Multiple ERG targets are likely to be directly or indirectly implicated in the control of coagulation. However, the gene target consistently regulated by ERG across different tissues and in vitro was the anticoagulant *TM*. Studies on the transcriptional regulation of TM have shown that KLF2 drives TM expression in

EC[20], particularly under high laminar SS conditions[21]. In this study, we show that ERG increases KLF2 binding to the TM promoter by facilitating chromatin access via p300 recruitment and H3K27 acetylation. We find that the ERG binding sites on the proximal TM promoter are essential for ERG and KLF2-dependent transactivation. These data suggest that both direct cooperation between ERG and KLF2 and epigenetic regulation controlled by ERG contribute to the fine regulation of TM

**Fig. 6** ERG cooperates with KLF2 to drive thrombomodulin expression. **a–d** TM promoter luciferase reporter assays. Values represent the fold change in relative luciferase activity over the empty pGL4 vector alone. **b** pcDNA or ERG expression plasmid (ERG) were co-transfected with a TM wild type (WT) promoter-luciferase construct (TM WT) in siCtrl or siKLF2-treated HUVEC, and luciferase activity was measured (n = 3 independent experiments). **b** pcDNA or KLF2 cDNA expression plasmid (KLF2) were co-transfected with TM WT promoter construct in siCtrl or siERG-treated HUVEC (n = 3 independent experiments). Graphical data presented in **a** and **b** are mean ± s.e.m., *P < 0.05, **P < 0.01, Student's t-test. **c** pcDNA, KLF2 or both KLF2 and ERG plasmids were co-transfected with TM WT promoter construct (n = 4 independent experiments). Graphical data are mean ± s.e.m., ***P < 0.001, ****P < 0.0001, One-way ANOVA, compared to pcDNA-TM WT condition. **d** pcDNA or KLF2 plasmid were co-transfected with TM promoter constructs (TM WT, TM mutant 1 or TM mutant 2) (n = 4 independent experiments). Graphical data are mean ± s.e.m., ***P < 0.001, ****P < 0.0001, One-way ANOVA. ##P < 0.01, compared to pcDNA-TM WT, Student's t-test. **e, f** ChIP-qPCR using primers to region R1 and R2 on **e** ERG- or **f** KLF2-bound chromatin from siCtrl, siERG or siKLF2-treated HUVEC (n = 3 independent experiments). Primers for a region within 5'UTR region of TM gene (TM neg con) were used as negative control. Data are shown as fold change over IgG. Graphical data are mean ± s.e.m., NS: Not Significant, *P < 0.05, **P < 0.01, ***P < 0.001, Student's t-test. **g** Representative image of proximity ligation assay (PLA) for nuclear ERG-KLF2 interaction was performed on confluent HUVEC using anti-ERG and anti-KLF2 antibodies; nuclei are identified by DAPI (blue). Scale bar, 20 μm. **h** Model: In homeostatic EC, ERG is required for p300 recruitment at the TM promoter region, leading to acetylation of H3K27 which opens the chromatin and allows KLF2 binding to the TM promoter. ERG binds to and cooperates with KLF2 to drive TM expression. In the absence of ERG, the chromatin is not accessible; KLF2 is not binding to the TM promoter and is not able to transactivate the TM promoter. Source data are provided as a Source Data file

expression in EC. A previous study reported cooperation between ERG and KLF2 in driving Flk1 expression in *Xenopus* embryo. Physical interaction between over-expressed ERG and KLF2 in a non-EC line was described[48], supporting our evidence of an endogenous ERG-KLF2 interaction in EC.

In this study, we report a mechanism of differential transcriptional control of TM expression depending on SS. We show that ERG is essential for chromatin remodelling and for KLF2-dependent regulation of TM selectively in low SS conditions in vitro. Our findings suggest that ERG, together with low levels of KLF2, provides the basal essential anticoagulant properties required throughout the endothelium of the microcirculation; however, extra protection from unwanted thrombus formation in a larger vessel, particularly arteries, is provided by pathways induced by high laminar SS. Indeed, high SS has been shown in vitro to promote endothelial epigenetic changes leading to an increase in open and accessible chromatin[47]. This could allow chromatin access for KLF2 and other TF to bind DNA and drive gene expression, without requiring ERG. In line with the concept of the additional KLF2-dependent protection, Boon et al.[49] have demonstrated that increased KLF2 expression in arterial high SS conditions results in activation of an endothelial athero-protective phenotype, by inhibiting the TGFβ-SMAD pathway. Interestingly, we have recently shown that ERG represses the TGFβ-SMAD pathway in EC to prevent endothelial-to-mesenchymal transition and fibrinogenesis[36]; whether ERG and KLF2 also cooperate in the regulation of the TGFβ pathway remains to be investigated. These studies suggest a model of cooperation between ERG and KLF2 in the transcriptional control of endothelial homeostasis, where ERG, partly by promoting KLF2 binding to DNA, is essential to maintain a basal, healthy and anti-thrombotic endothelium in the microvasculature of multiple organs. This pathway appears dispensable in blood vessels exposed to high SS, where the role of KLF2 becomes dominant, and possibly other ETS factors (such as ETS1[50]) or other epigenetic mechanisms may step in to provide cooperation. Future studies will investigate the TF networks involved in shear-dependent, cooperative transcriptional and epigenetic networks.

In vivo, endothelial deletion of KLF2 alone in adult mice does not cause a spontaneous thrombotic phenotype; double deletion of KLF2 and KLF4 is required to cause a profound compromise in vascular integrity and dysregulation of the coagulation system[26]. Crucially, deletion of endothelial ERG is sufficient to cause thrombosis and coagulopathy, with decreased platelet counts, decreased circulating levels of fibrinogen and increased D-dimer and TAT plasma concentrations. The penetrance and severity of the phenotype in *Erg*[iEC-KO] mice appears to be variable and vascular bed-specific. In fact, the thrombotic and hemorrhagic phenotypes observed in *Erg*[iEC-KO] mice occur selectively in the liver and the lung, showing intriguing tissue-specificity of this pathway. Our data also revealed an increased rate of thrombin generation in *Erg*[iEC-KO] mice and an ERG-dependent regulation of a set of genes potentially involved in the control of coagulation (such as *PAI1* and *eNOS*). Interestingly, acute treatment with a red blood cells-targeted TM was able to restore the levels of TAT, an early biomarker of coagulopathy and thrombosis, showing that loss of TM expression is in part responsible for the prothrombotic phenotype observed in ERG-deficient mice. Future experiments will investigate the contribution of other pathways in the prothrombotic phenotype of *Erg*[iEC-KO] mice.

Although regulation of TM expression is clearly a pivotal mechanism through which ERG exerts its anti-thrombotic function, it is likely that the thrombotic phenotype caused by inducible deletion of ERG results from a combination of pathways, which may have different relevance depending on tissue-specific mechanisms, still to be characterised. EC from different sites of the vascular tree appear indeed to use a site-specific combination of hemostatic proteins to prevent excessive clot formation[4]. Specifically, the liver synthesizes and releases most of the proteins involved in the clotting cascade (such as fibrinogen, protein C and protein S)[51]. This high local concentration of coagulation factors and the unique architecture of liver sinusoids, presenting fenestrated and discontinuous EC[52], might explain why the most pronounced phenotype in *Erg*[iEC-KO] mice is observed in this tissue. Moreover, one of ERG's key function is to promote vascular stability though the transcriptional control of genes, such as VE-cadherin and claudin-5[27]. Thus, it is possible that the hemorrhagic phenotype, observed in the lung of adult *Erg*[iEC-KO] mice, may be due to the combination of disrupted blood vessels with increased permeability and impairment of coagulation, leading to bleeding.

In conclusion, our study shows that ERG is crucial for the maintenance of the endothelial anti-thrombotic environment. We demonstrate that ERG is a direct transcriptional regulator of TM, driving its expression in a context-specific manner by cooperating with KLF2. The identification of vascular bed-specific pathways has potentially important implications for human pathologies, given that thrombotic diseases show selective preferences for specific organs' vascular beds. This pathway could be involved in the pathogenesis of human disease associated with thrombosis in microvessels and/or with coagulopathy, such as disseminated intravascular coagulation, hemolytic uremic syndrome or sepsis.

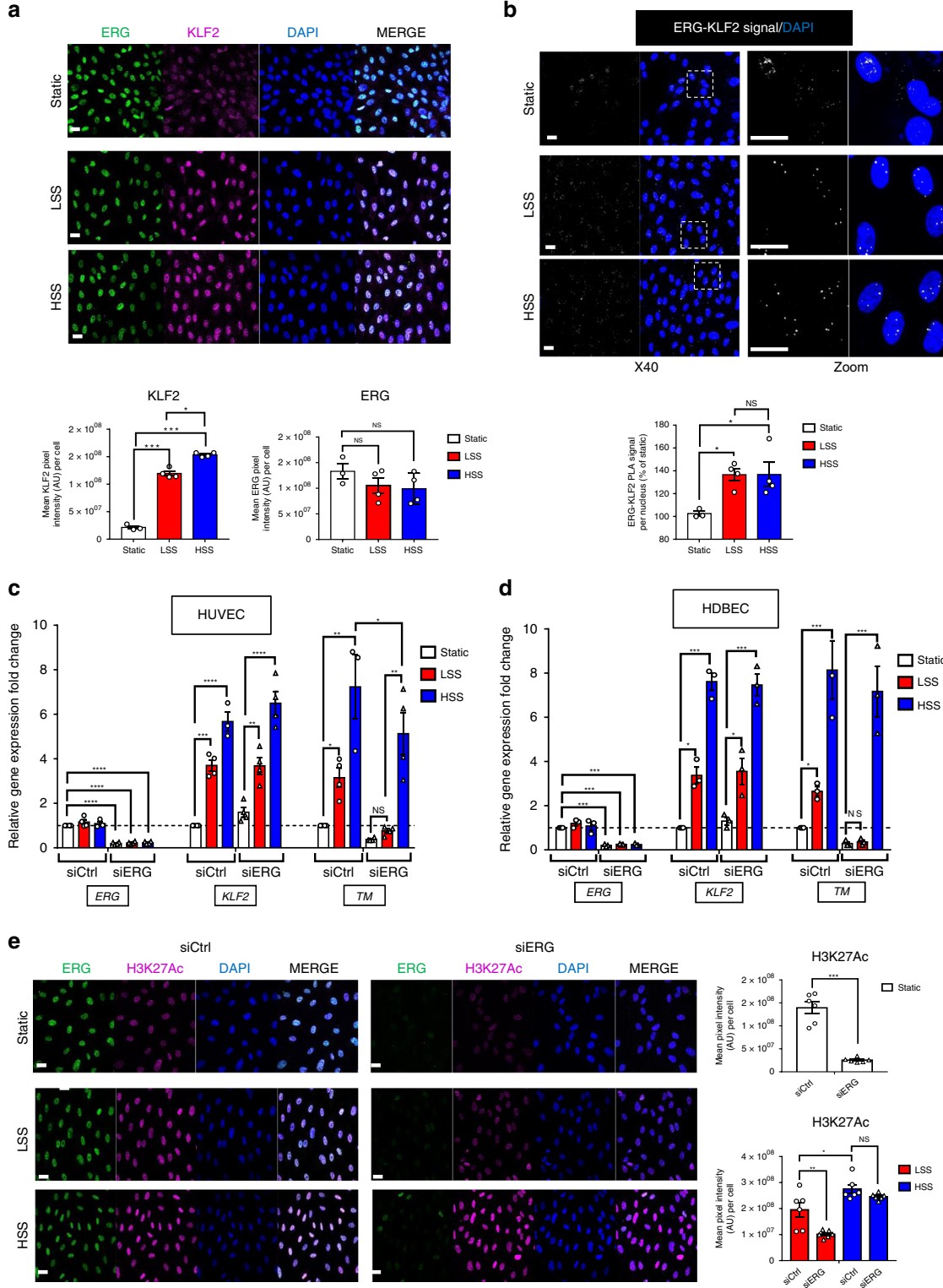

## Methods

**Mice and breeding**. All animal experiments were conducted according to and in compliance with the guidelines from Directive 2010/63/EU of the European Parliament for animal testing and research, with ethical approval from Imperial College London under UK Home Office Licence number PPL PEDBB/1586 and in compliance with the UK Animals (Scientific Procedures) Act of 1986. The inducible endothelial-specific ERG knockout mouse model ($Erg^{iEC-KO}$) was generated by breeding $Erg^{fl/fl}$ mice[30,33] with Cdh5(PAC)-CreERT2 mice[53]; 10–12 weeks old male and female mice were used for this study. Endothelial deletion of ERG was induced in adult mice, between 6–8 weeks, by tamoxifen injection (five injections of

0.5 mg daily). All experiments were conducted using littermate controls; mice were monitored, and tissues were harvested 25, 30 or 45 days post-tamoxifen injection.

**Cell culture and flow experiments**. Pooled human umbilical vein endothelial cells (HUVEC) were purchased from Lonza (C2519A) (Wokingham, United Kingdom) and cultured in Endothelial Cell Growth Medium-2 media (EGM-2) (Lonza) or isolated from umbilical cords[33,36] and cultured in M199 media. Human aortic endothelial cells (HAEC) (single donor, Promocell, C-12271) were cultured in EGM-2 media (Lonza). Human dermal microvascular endothelial cells (HDMEC)

**Fig. 7** ERG drives thrombomodulin expression in vitro selectively in low shear stress conditions. **a** Representative image and quantification of ERG (green) and KLF2 (magenta) expression in HUVEC cultured under static conditions or exposed to low shear stress (LSS, 5 dynes per cm$^2$) or high shear stress (HSS, 20 dynes per cm$^2$) for 24 h by immunofluorescence; nuclei are identified by DAPI (blue). Scale bar 40 μm. Quantification represents the mean pixel intensity for ERG or KLF2 signal (arbitrary unit, A.U.) per cell ($n = 3$–4 images per condition). **b** Representative image of proximity ligation assay (PLA) for nuclear ERG-KLF2 interaction was performed on confluent HUVEC cultured under static conditions or exposed to low shear stress (LSS, 5 dynes per cm$^2$) or high shear stress (HSS, 20 dynes per cm$^2$) for 24 h; nuclei are identified by DAPI (blue). Quantification represents the PLA signal (mean number of dots per cell) and is expressed as a percentage compared to static condition ($n = 3$–4 images per condition). Scale bar, 40 μm. **c** qPCR analysis of *ERG*, *KLF2* and *TM* mRNA expression in siCtrl or siERG-deficient HUVEC under static conditions or after 24 h under LSS or HSS ($n = 4$). **d** qPCR analysis of *ERG*, *KLF2* and *TM* mRNA expression in siCtrl or siERG-deficient HDBEC under static conditions or after 24 h under LSS or HSS ($n = 3$ independent experiments). **e** Representative image and quantification of H3K27Ac expression (magenta) in siCtrl or siERG-deficient HUVEC cultured under static conditions or exposed to LSS or HSS for 24 h by immunofluorescence; nuclei are identified by DAPI (blue). Scale bar 40 μm. Quantification represents the mean pixel intensity for H3K27Ac signal (arbitrary unit, A.U.) per cell ($n = 6$ images per condition). All graphical data are mean ± s.e.m., *$P < 0.05$, **$P < 0.01$, ***$P < 0.001$, Student's $t$-test for top panel and one-way Anova for bottom panel. Source data are provided as a Source Data file

(HDMEC comprise blood and lymphatic microvascular endothelial cells; single donor, Promocell, C-12212) and human dermal blood endothelial cells (HDBEC) (HDBEC are only blood microvascular EC which are positive for CD31 and VWF and negative for podoplanin; single donor, Promocell, C-12225) were cultured in Endothelial Cell Media MV2 (Promocell). All endothelial cells were transfected with plasmid or siRNA in EGM-2 media (Lonza). Cells were transfected with siRNA against ERG exon6 or exon7 (siRNA #2) (20 nM), both denoted as siERG in the text, and/or with siRNA against KLF2 (20 nM). An AllStars Negative Control siRNA (Qiagen) was used, which is denoted as siCtrl. Sequences of the siRNA are listed in Supplementary Table 1.

For flow experiments, transfected endothelial cells were re-plated after 24 h on μ-slides I$^{0.4}$ Luer-ibiTreat (ibidi) and placed under control static conditions, low (5 dynes per cm$^2$) or high laminar shear stress (20 dynes per cm$^2$) conditions for 24 hours using a parallel plate system (ibidi).

For some experiments, HUVEC were washed five times with starvation medium (EBM2 (Lonza) supplemented with 1%-Bovine Serum Albumin (BSA)) and treated with DMSO (Sigma) or C 646, p300/CBP Inhibitor (Abcam, ab142163) (5 or 10 μM) for 1 or 4 h.

**Immunoblotting analysis**. Immunoblotting was performed according to standard conditions and immunoblots were labelled with the primary antibodies listed in Table 1. Primary antibodies were detected using fluorescently labelled secondary antibodies: goat anti-rabbit IgG DyLight 680 and goat anti-mouse IgG Dylight 800 (Thermo Scientific). Detection of fluorescence intensity was performed using an Odyssey CLx imaging system (Li-COR Biosciences, Lincoln) and Odyssey ver.4 software. In some instances, primary antibodies were detected using HRP-conjugated secondary antibodies: anti-rabbit (Cell Signalling, 7074S), anti-mouse (GE HealthCare UK, NXA931V) and anti-goat (Dako, P0449). Fluorescence intensity or chemiluminescence were quantified by densitometry using Fiji-ImageJ software to measure protein levels and normalized against loading controls. The primary antibodies used are listed in Table 1 and in Supplementary Table 2. See source data file for the uncropped immunoblots.

**Immunofluorescence analysis of HUVEC and mouse tissues**. HUVEC for immunofluorescence and PLA were fixed with 4% paraformaldehyde for 15 minutes and permeabilized with 0.5% Triton before blocking with 3% BSA. PLA was performed according to the manufacturer's instructions using the Duolink In Situ Orange Kit Mouse/Rabbit (Sigma).

Tissues were fixed in 4% paraformaldehyde for 2 h and paraffin- (liver, lung) or OCT-embedded (aorta). Sections were stained for hematoxylin/eosin or Masson's Trichrome or incubated with primary antibodies for immunofluorescence staining. The primary antibodies used are listed in Table 1 and in Supplementary Table 2. Nuclei were visualized using DAPI (4′,6-diamidino-2-phenylindole). Confocal microscopy was carried out on a Carl Zeiss LSM780. Images were analysed with ImageJ (NIH) and Volocity (Version 6.3, PerkinElmer).

**Real-time polymerase chain reaction**. Total RNA from tissues and HUVEC was isolated by using the RNeasy kit (Qiagen) and reverse transcribed into cDNA using Superscript III Reverse Transcriptase (Invitrogen).

Quantitative real-time PCR was performed using PerfeCTa SYBR Green Fastmix (Quanta Biosciences) on a Bio-Rad CFX96 system. Gene expression values were normalized to GAPDH expression (human) or 18s (mouse). See Supplementary Tables 3 and 4 for the list of oligonucleotides.

**Plasmids**. Human ERG cDNA (NCBI Accession NM_182918) was cloned into the mammalian vector pcDNA3.1 (Invitrogen). PpyCAG-KLF2-IB was a gift from Austin Smith (Addgene plasmid # 60441; http://n2t.net/addgene:60441; RRID: Addgene_60441)[54]. A 1.078 kb region of the TM promoter proximal to the TSS was PCR amplified from human genomic DNA and cloned into the pGL4.10[luc2]

Luciferase Reporter Vector (Promega). See Supplementary Table 5 for oligonucleotide sequences. pGL4.10[luc2] (Promega, Madison, USA) Firefly Luciferase empty vector was used as a control. pGL4.74 [hRluc/TK] vector) (Promega) Renilla luciferase vector was used as an internal control.

**Multi site-directed mutagenesis**. Mutation of the ERG binding sites (EBS) on the thrombomodulin promoter luciferase construct (TM WT) was carried out using QuikChange Lightning Multi Site-Directed Mutagenesis Kit (Agilent, Berkshire, UK), according to the manufacturer's instructions. Two AGGAA motifs were changed to ACCAA to eliminate the ERG DNA binding motifs and generate the thrombomodulin mutant 1 promoter (TM mut 1). In addition to these two mutations, two other GGAA motifs were changed to CCAA and GCAA to generate the thrombomodulin mutant 2 promoter (TM mut 2). Sequences of the mutagenic oligonucleotides are listed in Supplementary Table 5.

**Reporter assays**. HUVEC were transfected with 1 μg of ERG plasmid or pcDNA3.1 empty vector and with luciferase reporter plasmids for 24 h using GeneJuice® Transfection Reagent (Merck). Luciferase activity was measured using the Dual-Luciferase Reporter Assay System (Promega) and a GloMax-Multi + Microplate Multimode Reader (Promega). Luciferase reporter activity was normalized to the internal Renilla luciferase control and is expressed relative to the pGL4 empty vector.

**Chromatin immunoprecipitation-qPCR**. Chromatin immunoprecipitation (ChIP) was performed using the ChIP-IT Express kit (Active Motif)[33], according to the manufacturer's instructions. In brief, HUVEC were cross-linked for 10 min with 1% formaldehyde. Chromatin was sheared using a Bioruptor UCD-200 ultrasound sonicator (Diagenode) and immunoprecipitated with 3 μg antibody to ERG (sc-354X, Santa Cruz Biotechnology), KLF2 (rabbit, ab203591, Abcam), H3K27Ac (rabbit, 39133, Active Motif) or p300 (mouse, ab14984, Abcam). The respective negative controls were rabbit IgG (PP64, Chemicon, Millipore) and mouse IgG (12- 371, Millipore). Immunoprecipitated DNA was then used as template for quantitative PCR using primers specific for genomic loci and listed in Supplementary Table 5.

**Bioinformatic analysis**. ERG ChIP-sequencing data shown in this study can be downloaded from the NCBI Gene Expression Omnibus (GEO) portal, accession number GSE124893[41]. ERG transcription factor motif discovery was performed using the JASPAR database (http://jaspar.genereg.net). Genome-wide ChIP-Seq data for H3K27Ac and H3K4Me3 histone modifications, RNA polymerase II occupancy in HUVEC and phyloP sequence conservation (plotted as conservation scores −5/+5) were obtained from the Broad Institute and publicly available from the ENCODE Consortium[40]. Tracks were visualized using the UCSC Genome Browser database (https://genome.ucsc.edu/index.html).

**Activated Protein C assay on HUVEC**. HUVEC transfected with siRNA as described above were re-seeded in a 24-well plate and Activated Protein C (APC) assay was performed after 48 h of siRNA treatment. Cells were washed with assay buffer (20 mM Tris, 100 mM NaCl, 1 mM CaCl$_2$, 0.1% BSA, pH 7.5) and incubated at 37 °C with human protein C (300 nM; Haematologic Technologies Inc. (HTI)) and human thrombin (10 nM; HTI). Samples were removed after 10, 20, 30 and 60 min and transferred to tubes containing human anti-thrombin III (100 nM; HTI) and heparin (15U per mL).

For quantification of the APC generated, samples (100 μl) were transferred to a 96-well plate and S-2366 substrate (0.5 mM, Chromogenix) was added to the wells. Cleavage of the substrate was measured at 405 nm for 20 min and the concentration of APC was calculated compared to a standard curve derived from pure APC with a known concentration (HTI). The concentrations were normalised to the concentration of total protein for each well and expressed as percentage compared to control siRNA following 60 min reaction time.

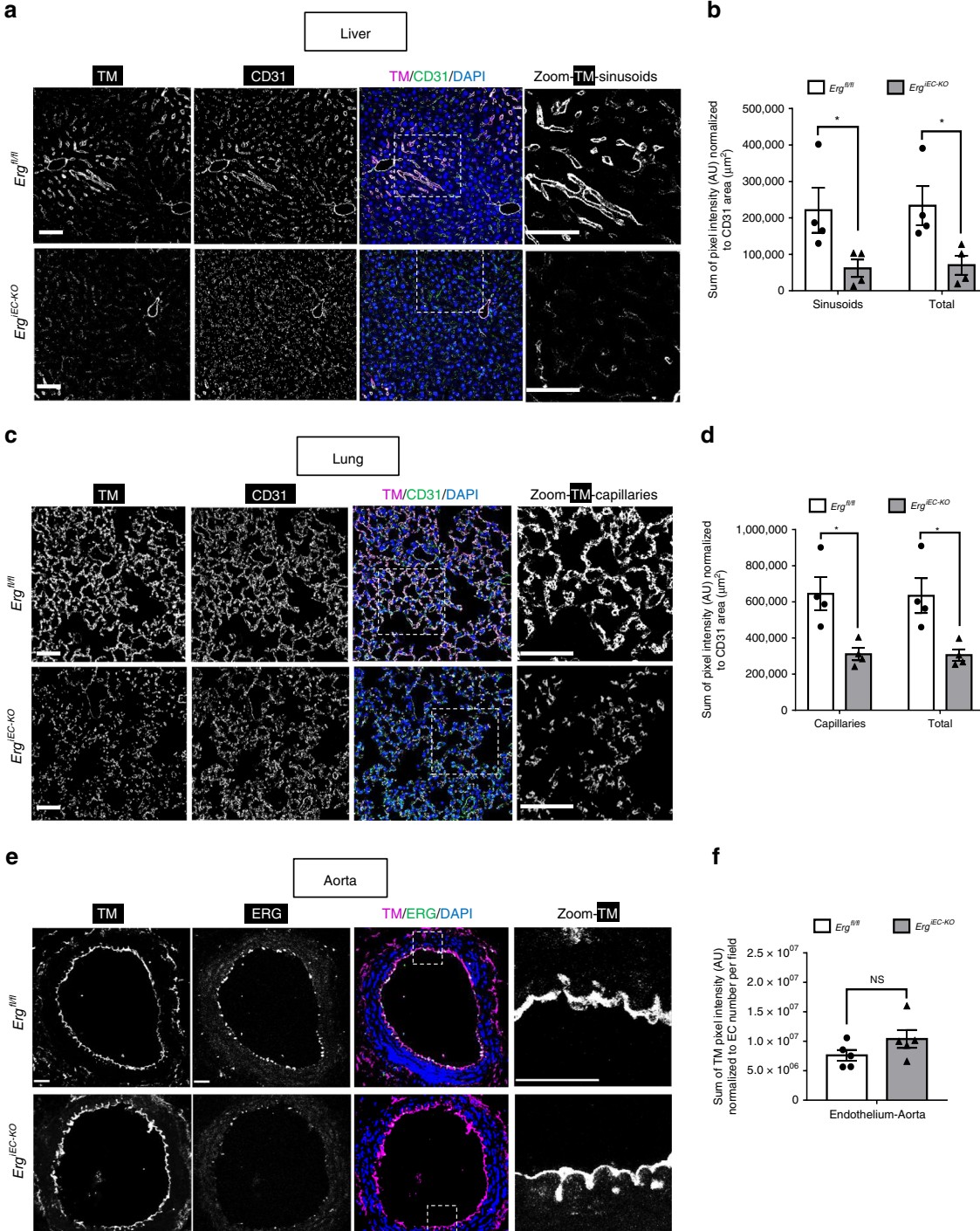

**Fig. 8** ERG regulates thrombomodulin expression specifically in microvascular beds. **a**, **c** Representative image by immunofluorescence and **b**, **d** quantification of TM expression (magenta) in **a**, **b** liver and **c**, **d** lung sections from $Erg^{fl/fl}$ and $Erg^{iEC-KO}$ mice 30 days after tamoxifen injection; tissues are co-stained for CD31 to visualise blood vessels (green); nuclei are identified by DAPI (blue). Scale bar 50 μm. Quantification represents the ratio between the sum of pixel intensity for TM signal (arbitrary unit, A.U.) and CD31 area ($μm^2$) (reflecting vascular coverage) ($n = 4$ mice per genotype). **e** Representative immunofluorescence image and **f** quantification of endothelial TM expression (magenta) in aorta sections from $Erg^{fl/fl}$ and $Erg^{iEC-KO}$ mice 45 days after tamoxifen injection; tissues are co-stained for ERG (green); nuclei are identified by DAPI (blue). Scale bar 50 μm. Quantification represents the ratio between the sum of pixel intensity for TM signal (arbitrary unit, A.U.) and the number of EC per field identified by DAPI ($n = 5$ mice per genotype). All graphical data are mean ± s.e.m., *$P < 0.05$, **$P < 0.01$, ***$P < 0.001$, Student's $t$-test. Source data are provided as a Source Data file

**Hematologic analysis**. Mice were anesthetised intraperitoneally with keta-mine (75 mg per kg)/medetomidine (1 mg per kg) and bled from the retro-orbital plexus into tubes containing 3.8% citrate anticoagulant. Platelet counts were evaluated by flow cytometry using calibrated beads (Saxon Europe) and rat anti-mouse GPIbβ-DyLight 488 (X488; Emfret) according to the

manufacturer's instructions. Briefly, whole blood was diluted 1/20 with modified Tyrode's buffer and platelets were stained with rat anti-mouse GPIbβ-DyLight 488 for 15 min at room temperature. Samples were analysed using a FACSCalibur/LSRFortessa™ flow cytometer and Flowing Software (v2.5.1).

**ELISA**. Hemolytic plasma samples were excluded. Plasma levels of D-dimer, Thrombin-antithrombin (TAT) and fibrinogen, were respectively determined using a mouse D-dimer-ELISA kit (Clone Corp), a mouse TAT-ELISA kit (Abcam) and a mouse fibrinogen ELISA kit (Abcam) according to the manufacturer's instructions. Samples were run in duplicate with $n = 3$–8 animals per group.

**Thrombin generation assay**. Thrombin generation in citrated mouse plasma was assessed by calibrated automated thrombography (CAT)[55,56]. Briefly, single mouse plasma preparations (40 µl per well) were used with 1 pM tissue factor (Dade Innovin, Dade Behring), 4 µM phospholipid vesicles and 16.6 mM $CaCl_2$. Contact activation coagulation was inhibited by adding corn 2 trypsin inhibitor (65 µg per ml plasma) and thrombin generation quantified by adding 0.42 mM of Z-GlyArgAMC-HCl (Bachem). Samples were run in duplicate with $n \geq 4$ animals per group.

**Production of mouse red blood cells-targeted thrombomodulin**. Fusion protein targeted to mouse glycophorin A was produced as described below[39]. Briefly, a stable Drosophila S2 cell lines expressing red blood cells-targeted thrombomodulin (RBC-TM) was maintained in Schneider's medium (Thermo Fisher Scientific) and transitioned to Insect-Xpress (Lonza, Walkersville, MD) for protein production. Supernatants were harvested after induction with $CuSO_4$. Protein also had a triple FLAG and was purified using anti-FLAG (M2) affinity resin (Sigma-Aldrich) and tested for purity via SDS-PAGE and HPLC (Waters, Huntingdon Valley, PA, USA) using a Yarra™ 3 µm SEC-2000 LC Column 300 × 4.6 mm size exclusion column with a flow rate of 0.35 mL per min (Phenomenex, Torrance, CA, USA). RBC-TM migrated on SDS-PAGE as a single approximately 85 kDa band.

**Characterization of RBC-TM in vitro**. Characterization of the antithrombotic properties of RBC-TM was assessed by performing APC generation assay in vitro. Free RBC-TM (4 or 40 µg per ml) was incubated with 5 nM bovine thrombin (Sigma-Aldrich, St Louis, MO) and 300 nM human protein C (Sigma-Aldrich, St Louis, MO) for 20 min at room temperature. Thrombin was quenched with 50 U per mL hirudin (Sigma-Aldrich, St Louis, MO), and APC was measured using APC substrate S-2366 (Diapharma, West Chester, OH, USA). All samples were run in quadruplicates. The rate of substrate hydrolysis was measured by monitoring the change in absorbance at 405 nm over time at room temperature using a Spectramax M2 Microplate reader (Molecular Devices, CA, USA). Data presented as maximum signal reached at 20 min.

**Acute treatment with RBC-TM in vivo**. Endothelial deletion of ERG was induced in adult mice (6 weeks) by tamoxifen injection. Twenty-four days post-tamoxifen injection, mice were bled via the tail vein for analysis of hemostasis-related plasma biomarkers (TAT, D-dimer and fibrinogen). The following day (25 days post tamoxifen), mice were intravenously (tail vein) injected with RBC-TM (4 mg per kg) and sacrificed 6 hours after this acute treatment. Mice were bled from the retro-orbital plexus for plasma markers analysis and tissues were collected for histology.

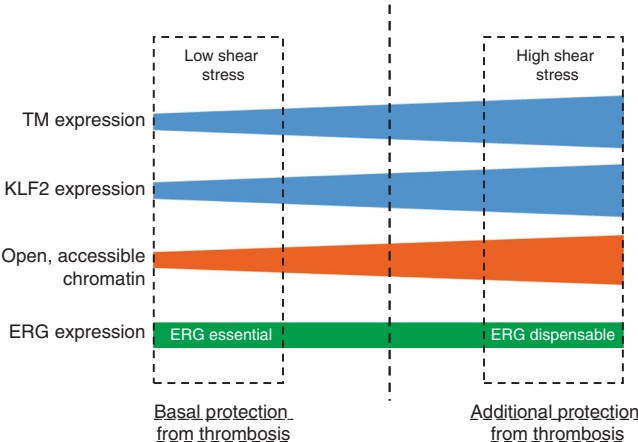

**Fig. 9** Distinct anti-thrombotic pathways in low versus high shear stress conditions. Model of vascular bed-specific anticoagulant pathway effective selectively in blood vessels exposed to low SS. Under low shear stress (SS), EC express relatively low levels of KLF2 and TM. ERG is essential for p300 recruitment and basal acetylation of histone H3 (H3K27Ac), a marker of open and accessible chromatin, and for KLF2 binding to the TM promoter, and thus is required for basal TM expression in EC. The cooperativity between ERG and KLF2 in driving TM expression maintains a basal anti-thrombotic environment selectively in microvasculature beds exposed to low SS. In contrast, high laminar SS mediates epigenetic modifications, including increased H3K27Ac levels, through different pathways, leading to an increase in open and accessible chromatin allowing transcription factors binding. KLF2 expression is also induced in this context and drives increased levels of TM, providing an additional protective mechanism in blood vessels (aorta and arteries) where thrombosis can be fatal. ERG is dispensable for this regulation; other transcription factors may cooperate with KLF2 to drive TM level in high shear stress conditions

**Statistical analysis**. All results presented in this study are representative of at least three independent experiments. '$n$' represents the number of biological replicates unless otherwise stated. Data are shown as the mean ± standard error of the mean (s.e.m.). Statistical significance was determined by two-tailed Student's $t$ test or one-way ANOVA with Tukey post hoc test using Prism ver. 6.0 (GraphPad). Differences were considered statistically significant with a $P$ value < 0.05.

**Reporting summary**. Further information on research design is available in the Nature Research Reporting Summary linked to this article.

## Data availability
Data supporting the findings of this work are available within the paper and its Supplementary Information file. The source data (included uncropped immunoblots) underlying Figs. 1f–i, 2a–f, 3a–e, 4a–g, 5b–f, 6a–f, 7a–e, 8b, d, f and Supplementary

**Table 1 Complete list of antibodies used for this study**

| Antibody (host) | Company | Catalogue number | Application and dilution |
| --- | --- | --- | --- |
| CD31 (rabbit) | Abcam | ab28364 | IF (mouse tissue, 1/200) |
| CD41 [MWReg30] (rat) | Abcam | ab33661 | IF (mouse tissue, 1/200) |
| Isolectin B4 (IB4) | Vector | FL-1201 | IF (mouse tissue,1/200) |
| Wheat Germ Agglutinin (WGA) | Thermo Fisher Scientific | W11261 | IF (mouse tissue, 1/400) |
| ERG (mouse) | Santa Cruz | sc-376293 | PLA (HUVEC, 1/200)/WB (HUVEC, 1/2000) |
| ERG (rabbit) | Abcam | ab133264 | IF (HUVEC, 1/200)/WB (HUVEC, 1/1000) |
| ERG (rabbit) | Abcam | ab92513 | IF (mouse tissue, 1/200)/WB (mouse tissue, 1/1000) |
| Fibrinogen (rabbit) | Abcam | ab34269 | IF (mouse tissue, 1/200) |
| GAPDH (mouse) | Millipore | MAB374 | WB (HUVEC, mouse tissue, 1/10000) |
| H3K27Ac (rabbit) | Active Motif | 39133 | IF (HUVEC, 1/200) |
| KLF2 (rabbit) | Abcam | ab203591 | PLA (HUVEC, 1/100)/WB (HUVEC, 1/500) |
| Actin, α-smooth muscle-Cy3 (mouse) | Sigma | C6198 | IF (mouse tissue, 1/400) |
| TM (mouse) | Dako | Clone 1009 | IF (HUVEC, 1/100) |
| TM (mouse) | Santa Cruz | sc-13164 | WB (HUVEC, 1/1000) |
| TM (goat) | R&D | AF3894 | IF (mouse tissue, 1/200)/WB (mouse tissue, 1/1000) |

List of antibodies used for immunofluorescence (IF), western blot (WB) and proximity ligation assay (PLA). Dilution of the antibodies used for each specific application is specified in the right column (in brackets)

Figs. 1a, b, d–h, 3a, 4a–h, 7a, b, 8a, c–h, 9c and 10 are provided as a Source Data file. Any other data are available from the corresponding authors upon reasonable request.

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

## Acknowledgements

We thank Dr Joseph J. Boyle, Prof Dorian Haskard and Prof Mike Laffan (Imperial College London, London, UK) for helpful discussions. This work was funded by grants from the British Heart Foundation (RG/11/17/29256 and RG/17/4/32662). Production and validation of the RBC-TM construct was funded by grants from National Institutes of Health (1R01 HL 128398-04 and 5T32HL007971-18).

## Author contributions

C. Peghaire conceived, designed and carried out experiments, analysed and conceptualised results, wrote the paper. N.P.D., M.L., I.I.S.-C. and J.A. designed and performed experiments, analysed and interpreted results, contributed to scientific discussion. V.K., C.R., C.P. performed experiments and analysed results, contributed to scientific discussion. L.I. performed experiments. R.K. and V.R.M. produced, validated and provided the mouse RBC-TM fusion protein. J.C.M. and G.M.B. contributed to scientific discussion. A.M.R. provided funding, conceived, designed and supervised the study, interpreted results and wrote the paper.

## Competing interests

The authors declare no competing interests.
