## [Peer Review File · Nature Communications]

Reviewers' comments:

Reviewer #1 (Remarks to the Author):

This manuscript by Peghaire and colleagues shows that ERG is required for baseline expression of thrombomodulin and that its loss in endothelial cells induces thrombosis. The study is well performed and the results are consistent, but the study is incrementally novel.

Major comments:

1. It is known that shear stress and KLF2 induce TM expression (Lin et al 2005, Dekker et al 2006, etc). That ERG is involved is a novel aspect, but that KLF2 and ERG interact has been described in the context of vascular development (Meadows et al 2009). The added news value of this manuscript is therefore below the standard of a Nat Comms paper.
2. The authors suggest that ERG is essential for expression of TM in low shear regions, but not in high shear regions. For the in vivo experiments, shear stress is very hard to determine, especially in smaller vessels, so it cannot be concluded that a potential difference in shear stress is responsible for the differential TM expression in large vs small vessels. The in vitro shear stress experiments are performed with HUVECs and this may not reflect the vascular beds studies in vivo.
3. The authors suggest, but do not show whether P300 is involved in acetylation of H3K27 in the TM promoter and the activation of TM expression. Can this be experimentally verified?
4. The PLA experiment is intriguing, but misses essential controls. Does the signal increase after shear stress stimulation? Does siERG or siKLF2 decrease the signal? Can this be quantified?
5. All experiments where the authors perform ChIP with ERG antibodies in control or siERG cells cannot be used to interpret ERG binding. If you reduce ERG levels in the cell using siRNA, you naturally also IP less ERG and less potential background signal, so these experiments do not show specific ERG binding to those promoter regions.

Minor comment:

Please confirm ERG overexpression after transfection by Western blot.

Reviewer #2 (Remarks to the Author):

In summary, the cooperation of ERG and KLF2 in SS-dependent regulation of TM in cultured cells is well documented and provides new insights in the flow-regulated TF network in EC. The key finding of this work is that "ERGtogether with low levels of KLF2.....is essential for the regulation of TM

under low SS conditions (basal) but dispensable in the context of high SS". This notion is well supported by in vitro data in HUVECS, using an array of different experiments to show physical interaction of ERG and KLF2, occupancy of the TM promoter via CHIP, as well as gain and loss of function transfection experiments. In vivo data from iEC-ERG KO mice show that TM gene expression is suppressed in lung and liver.

The In vivo data in support of a tissue/organ-selectivity activity of the ERG-KLF2 interaction, and a causal effect of altered TM gene expression on the observed phenotype of thrombosis are insufficient, especially with regard to the pleiotropic effects of ERG deletion in endothelium.

Three Major issues are:

A weaker aspect of the work is the documentation of thrombosis: ERG-KO mice also show bleeding in the lung and liver, but not in brain or kidney; this phenotype was associated with plasma indicators of a procoagulant state., such as slightly elevated DDimer, moderate platelet consumption and somewhat increased peak thrombin generation in CAT assay, but unaltered ETP:

- The presence of IHC-reactive fibrinogen is insufficient to document thrombosis: this requires demonstration of platelet aggregates and/or fibrin, but the AB likely does not differentiate between fibrin and fibrinogen.
- The "progressive" nature of thrombosis is insufficiently documented (N=3/4 versus 4/4)
- The procoagulant state should be further characterized by measuring at least prothrombin and fibrinogen levels, as these are also acute phase reagents
- Ubiquitous, conditional TM gene ablation in adult animals was not reported to result in overt thrombosis [PMID 28920104]

Data supporting the tissue/organ-selectivity of ERG-KLF-dependent TM expression and thrombosis are insufficient:

- TM expression is only shown for lung and liver.
- Evidence for thrombosis is marginal (see above)
- The cause-effect relation between hemorrhage and coagulation remains unclear. Extravascular blood, or blood in contact with dysfunctional or partially disrupted endothelium tends

indeed to clot. I would encourage the authors to consult reference 26 as an example for a thoughtful evaluation of thrombosis and hemorrhage controlled by KLF2 and KLF4.

A general and strong concern regarding overall data interpretation (especially in vivo observations) is -as correctly pointed out by authors in the discussion p. 25- that conditional ablation of ERG in EC has pleiotropic effects, including lung and liver, that may largely be responsible for the observed phenotype. To establish a cause-effect relation between ERG-KLF2 interaction and thrombosis, it might be necessary to genetically remove the ERG-binding region in the proximal TM promoter that was detected in the CHIP assays and measure TM expression and thrombosis/fibrin deposition in different organs.

Minor issues are:

- Page 6/results: The reference cited in results for the iEC-model (reference 36) used PDGFB-eGFP-Cre, while the methods state that Cdh5(PAC)-CreERT2 was used. Please clarify which one was used, and whether PDGFB- and Cdh5 Cre have different phenotypes?
- Figure 4 a: the resolution of this image is insufficient to provide meaningful information (such as the sequence data provided in supplemental data).

Reviewer #3 (Remarks to the Author):

This manuscript, Peghaire et al have identified an additional role for the endothelial ETS transcription factor ERG in vascular specific regulation of thrombosis. Along with its role in early endothelial cell development and angiogenesis, ERG has also been implicated in the maintenance of vascular integrity postnatally. Here, the authors describe how ERG, through its regulation of the anti-thrombotic receptor, Thrombomodulin, may prevent thrombosis during low shear stress conditions in vitro and in vivo.

In general, the data presented in the manuscript are detailed and rigorous and mostly support the overall conclusion of the study. The direct regulation of the TM promoter by ERG needs to be

bolstered, and the direct relationship between ERG and KLF in the regulation of the TM promoter is not convincing. The authors should address my concerns, indicated below, prior to publication of this work.

Main concerns:

1) The authors conclude that Erg directly regulates TM via the ETS binding sites in the TM promoter. To demonstrate this more definitively, they should mutate the ETS binding sites in the TM promoter to determine if mutation of those sites results in a loss of Erg-dependent transactivation. As it is, the ChIP data provide circumstantial evidence for a direct, functional interaction, but otherwise the results could be interpreted as an indirect regulation of TM by Erg and there is no evidence presented that Erg binding is functional.

2) Likewise, the conclusion that Erg recruits KLF to the TM promoter is not well supported. The proximity ligation assay demonstrates that the two factors are closely situated on chromatin but not necessarily in contact, and definitely do not support the idea that Erg recruits KLF to the promoter. The data are more simply explained by the increase in H3K27Ac that is mediated by Erg, as shown by the authors' own data. Indeed, as the authors show, in the presence of high shear stress, Erg is not required for KLF binding, presumably because of increase H3K27Ac at TM caused by high shear.

Therefore, to strengthen their results regarding the cooperative action of ERG and KLF in regulating the TM promoter, the authors should perform an additional transactivation assay to test whether the combined expression of ERG and KLF2 results in cooperative activation of the TM promoter cooperatively. Additionally, to supplement their ChIP-PCR experiment, the authors can also use the mutant TM promoter (mERG) from the previous point to determine if KLF transactivates the TM promoter in the absence of Erg binding sites.

4) In the presence of high shear forces, in the presence and absence of Erg, does H3K27Ac at the TM promoter increase? This needs to be tested to provide some mechanistic insight into the requirement of Erg for TM expression and subsequent regulation of thrombosis.

5) (Minor) In Figure 4A, it would be useful to show the entire Thrombomodulin locus, as well as the zoomed view of the region of interest.

Reply to reviewers: Manuscript NCOMMS-18-24141A

Low shear stress-dependent regulation of thrombosis in the microvasculature orchestrated by the transcription factor ERG

Reviewers' comments:

Reviewer #1 (Remarks to the Author): *We thank the Reviewer for his/her comments*

This manuscript by Peghaire and colleagues shows that ERG is required for baseline expression of thrombomodulin and that its loss in endothelial cells induces thrombosis. The study is well performed and the results are consistent, but the study is incrementally novel.

Major comments:

1. It is known that shear stress and KLF2 induce TM expression (Lin et al 2005, Dekker et al 2006, etc). That ERG is involved is a novel aspect, but that KLF2 and ERG interact has been described in the context of vascular development (Meadows et al 2009). The added news value of this manuscript is therefore below the standard of a Nat Comms paper.

The interaction between recombinant, over-expressed KLF2 and ERG in a non-endothelial system has indeed been reported. However, our study takes this much further. We show that endogenous ERG and KLF2 form a nuclear transcriptional complex in human endothelial cells, i.e. in the context where these two TF are more relevant. Moreover, the study by Meadows et al. was carried out in the context of vascular development, in a Xenopus model. The physiological context of our study investigates the regulation of thrombomodulin (TM) expression and thrombosis in adult mouse. We disagree with the view that our study is not novel, because apart from this previous report of KLF2-ERG proteins interaction, nothing was known about its physiological significance in the regulation of endothelial gene expression, and the functional interdependence of the two transcription factors. The substantial amount of data in our study provides new significant insights into the molecular and transcriptional mechanisms that regulate endothelial homeostasis and thrombosis during adulthood in different vascular beds.

2. The authors suggest that ERG is essential for expression of TM in low shear regions, but not in high shear regions. For the in vivo experiments, shear stress is very hard to determine, especially in smaller vessels, so it cannot be concluded that a potential difference in shear stress is responsible for the differential TM expression in large vs small vessels. The in vitro shear stress experiments are performed with HUVECs and this may not reflect the vascular beds studies in vivo.

We agree with the Reviewer that measurement of flow and shear stress (SS) in vivo is challenging (especially for small vessels). Although the literature reports different models (human, mouse, rat, pig) and different methods (rheology, micro-CT, computational fluid dynamics and modelling) leading to variable measurements, there is evidence that some microvascular beds are exposed to low flow and shear stress (liver, lung, spleen)^{1,2,3}. More importantly, many studies have demonstrated that aorta and large arteries are exposed to high SS^{4,5}.

It is well known that TM and KLF2 are key flow/shear stress responsive genes. Our in vitro findings show an ERG-KLF2 cooperation in driving TM expression selectively under low SS in vitro. The evidence combined points to a crucial role of shear stress in this regulation.

To address the Reviewer's comment regarding the in vitro SS experiments in HUVEC, we performed additional flow experiments on endothelial cells isolated from different vascular beds, namely dermal microvascular endothelial cells (HDBEC) and human aortic endothelial cells (HAEC).

The results obtained in microvascular EC (HDBEC) (Fig. 7d) are similar to those obtained in HUVEC (Fig. 7c) and show that in these cells ERG is required for the regulation of TM only in low SS conditions but is dispensable under high SS.

However, the experiments in human aortic endothelial cells (HAEC) showed a different picture. In these cells, ERG is not required for TM expression, either in low or high SS conditions. The data is presented in Supplementary Fig. 10. The results are in line with the in vivo data on mouse aorta from ERG-deficient mice, shown in Fig. 8e-f, and indicate the presence of distinct molecular mechanisms of gene regulation in EC from different vascular beds. Interestingly, these differences are retained ex vivo, possibly via epigenetic memory of the cells being exposed to high SS in vivo. We thank the Reviewer for the suggestion which resulted in unexpected, exciting data.

3. The authors suggest, but do not show whether P300 is involved in acetylation of H3K27 in the TM promoter and the activation of TM expression. Can this be experimentally verified?

The acetyltransferase p300 has been shown to bind to TM promoter and to promote its expression⁶. Moreover, we have recently shown that ERG is involved in p300 recruitment and acetylation of H3K27 on super-enhancers associated with endothelial cells-enriched genes⁷.

To confirm that ERG is involved in p300 recruitment and acetylation of H3K27 at the TM promoter, we conducted a new set of experiments. Pharmacological inhibition of p300 led to a decrease in H3K27Ac occupancy on the TM promoter (Fig. 5d and Supplementary Fig. 7a), accompanied by a decrease in TM mRNA levels (Supplementary Fig. 7b), as expected. Moreover, our new ChIP-qPCR data presented in Fig. 5e shows that deletion of ERG in HUVEC led to a significant decrease in p300 occupancy associated with decreased H3K27Ac levels (Fig. 5f) on the TM promoter. These data demonstrate that ERG is required for p300 recruitment, leading to the acetylation of H3K27, opening of the chromatin on the TM promoter region and allowing transcription.

4. The PLA experiment is intriguing, but misses essential controls. Does the signal increase after shear stress stimulation? Does siERG or siKLF2 decrease the signal? Can this be quantified?

We appreciate the Reviewer's comment and performed additional controls for the PLA experiments. We repeated the PLA in HUVEC treated with control, ERG, KLF2 or ERG+KLF2 siRNA and quantified the PLA signal for each condition. The experiments confirmed the specificity of the PLA signal since ERG, KLF2 or ERG+KLF2 siRNA treatment were able to abolish this interaction (Supplementary Fig.9c).

We also performed new PLA experiments (Fig. 7a-b) on HUVEC cultured under SS conditions (low or high SS) as suggested by the Reviewer. We found that the ERG-KLF2 complex formation was increased in low SS compared to static conditions (Fig. 7b). Intriguingly, ERG-KLF2 complex was not further increased under high SS (Fig. 7b), suggesting that other partners/pathways support KLF2-dependent transcription in these conditions.

5. All experiments where the authors perform ChIP with ERG antibodies in control or siERG cells cannot be used to interpret ERG binding. If you reduce ERG levels in the cell using siRNA, you

naturally also IP less ERG and less potential background signal, so these experiments do not show specific ERG binding to those promoter regions.

Apologies for the lack of clarity. We carried out multiple controls for the ChIP-qPCR experiments (Fig. 5b). We used siERG-treated cells to check that the signal obtained with the ERG antibody is specific for ERG, because it should be absent in siERG-treated cells. IgG (now labelled on all graphs), raised from the same species as the ERG antibody (rabbit), was used for the ChIP experiments to determine the background and the non-specific signal on selected DNA regions. To check ERG binding on TM locus, qPCR was performed for 2 regions of the TM promoter (R1 and R2) that contain ERG binding motifs and overlap with our independent ERG ChIP-sequencing data⁷, but also on a negative region that does not contain any ERG binding sites (TM neg region). As expected, ERG binds specifically to regions R1 and R2 but not to the TM neg region.

Minor comment:

Please confirm ERG overexpression after transfection by Western blot.

We are happy to comply with reviewer request; we have added the Western blot data in Supplementary Fig. 4f to confirm ERG overexpression at the protein level in HUVEC.

Reviewer #2 (Remarks to the Author): *We thank the Reviewer for his/her comments*

In summary, the cooperation of ERG and KLF2 in SS-dependent regulation of TM in cultured cells is well documented and provides new insights in the flow-regulated TF network in EC. The key finding of this work is that “ERGtogether with low levels of KLF2.....is essential for the regulation of TM under low SS conditions (basal) but dispensable in the context of high SS”. This notion is well supported by in vitro data in HUVECS, using an array of different experiments to show physical interaction of ERG and KLF2, occupancy of the TM promoter via CHIP, as well as gain and loss of function transfection experiments. In vivo data from iEC-ERG KO mice show that TM gene expression is suppressed in lung and liver.

The In vivo data in support of a tissue/organ-selectivity activity of the ERG-KLF2 interaction, and a causal effect of altered TM gene expression on the observed phenotype of thrombosis are insufficient, especially with regard to the pleiotropic effects of ERG deletion in endothelium.

Three Major issues are:

A weaker aspect of the work is the documentation of thrombosis: ERG-KO mice also show bleeding in the lung and liver, but not in brain or kidney; this phenotype was associated with plasma indicators of a procoagulant state., such as slightly elevated DDimer, moderate platelet consumption and somewhat increased peak thrombin generation in CAT assay, but unaltered ETP:

- The presence of IHC-reactive fibrinogen is insufficient to document thrombosis: this requires demonstration of platelet aggregates and/or fibrin, but the AB likely does not differentiate between fibrin and fibrinogen.

We followed the Reviewer's advice and performed additional immunostaining for platelet using a CD41 antibody. The data presented in Fig.1e reveals the presence of several platelet aggregates in liver sinusoids of Erg^{iEC-KO} mice (30 days post-tamoxifen injection) and not in the control mice, in line with fibrinogen staining showing clots in hepatic sinusoids.

With regards to the fibrinogen antibody used in this study, the antibody does indeed recognize both mouse fibrinogen and fibrin. However, labelled fibrinogen and fibrin antibody have been used as tools in a laser-induced thrombosis model and have shown similar results in terms of visualisation and quantification of clots in vivo⁸, showing that fibrinogen can be used to investigate the presence of clots.

Therefore, the combination of these data is in our view convincing evidence of thrombosis in these vessels. To further strengthen the data, we have also carried out more in vivo experiments to increase the number of mice (see reply below).

- The “progressive” nature of thrombosis is insufficiently documented (N=3/4 versus 4/4)

This is a fair point which we have discussed at length amongst co-authors; it is difficult to determine the dynamics of coagulopathy over time. In the original submission, the in vivo data on thrombosis was observed in mice at day 30 and 45 after tamoxifen treatment. However we cannot definitely prove that this is a progressive phenotype, therefore we have removed the word “progressive” from the manuscript.

Moreover, we have carried out further experiments to increase the number of mice analysed at day 30 after tamoxifen treatment and strengthen the evidence of thrombosis and coagulopathy:

- *Histology: n=9 mice per genotype (combining previous and new data), 30 days post tamoxifen (new Supplementary Fig. 2a). The new data confirmed the presence of clots and/or hemorrhages in liver and the presence or not of hemorrhages in lung, confirming the variability of the thrombotic phenotype of Erg^{iEC-KO} mice.*
- *Platelet counts: n=8 mice per genotype (new data), 30 days post tamoxifen (Fig. 1f). The new data confirmed that ERG-deficient mice have lower platelet count.*
- *D dimer ELISA: n=8 mice per genotype (combining previous and new data), 30 days post tamoxifen (Fig. 1h). The new data confirmed elevated D-dimer plasma levels in Erg^{iEC-KO} mice compared to control mice.*

- The procoagulant state should be further characterized by measuring at least prothrombin and fibrinogen levels, as these are also acute phase reagents

We were happy to comply with the Reviewer’s suggestion and we performed fibrinogen and thrombin-antithrombin (TAT) ELISA on mouse plasma. We decided to measure TAT and not prothrombin levels since TAT ELISA is more sensitive but also provides information on thrombin activation/inactivation.

- *Fibrinogen ELISA: n=8 mice per genotype, 30 days post tamoxifen (Fig. 1g)*
- *TAT ELISA: n=8 mice per genotype, 30 days post tamoxifen (Fig. 1i)*

These new data show reduced fibrinogen levels and increased TAT levels in Erg^{iEC-KO} mice (in line with increased D-dimer levels), which are all markers of coagulopathy.

The modified table presented in new Supplementary Fig. 2a summarizes the in vivo phenotype 30 days post tamoxifen, including all the new data.

- Ubiquitous, conditional TM gene ablation in adult animals was not reported to result in overt thrombosis [PMID 28920104]

This is an interesting point; we have reviewed the paper by van Mans et al in detail and do not think that this paper is in contradiction with our study. In a previous paper⁹, the same group demonstrated that 60% of the mice with genetic constitutive deletion of TM only in EC (Tie2Cre driver line) die postnatally due to spontaneous thrombosis, resulting in consumption of coagulation factors, coagulopathy and generalized bleeding. In the van Mans paper, the authors showed that inducible global deletion of thrombomodulin (TM) in adult mice (8-14 weeks), using an ERCre driver line, led to a less marked thrombotic phenotype compared to the Tie2Cre mice. A small percentage of the ERCre, TMlox/lox mice died 2 weeks post-tamoxifen injection (the cause of the death has not been elucidated) and the majority of these mice develop thrombotic events mainly restricted to the tail and hind limb, showing that these mice do have a phenotype but with a variable penetrance. Notably, the ERCre, TMlox/lox mice show variable deletion efficiency in the lung. These differences make the direct comparison between this study and ours difficult to interpret.

Data supporting the tissue/organ-selectivity of ERG-KLF-dependent TM expression and thrombosis are insufficient:

- TM expression is only shown for lung and liver.

For the in vivo profiling of TM expression, we decided to focus our study on liver (Fig. 2d and Fig. 8a-b) and lung (Fig. 2e and Fig. 8c-d) since the prothrombotic and/or hemorrhagic phenotypes of Erg^{IEC-KO} mice are particularly evident in these tissues.

In order to validate our in vitro findings showing that ERG regulates TM expression specifically in low SS (microvasculature of liver and lung) but not in high SS conditions, we also carried out immunostaining on mouse aorta (Fig. 8e-f and Supplementary Fig. 10a-b) which is exposed to high SS^{4,5}. This analysis showed no regulation of TM expression by ERG in the aorta, in line with the in vitro model.

- Evidence for thrombosis is marginal (see above)

We hope that all the new data discussed above has persuaded this Reviewer that there is a thrombotic phenotype in the Erg^{IEC-KO} mice.

- The cause-effect relation between hemorrhage and coagulation remains unclear. Extravascular blood, or blood in contact with dysfunctional or partially disrupted endothelium tends indeed to clot. I would encourage the authors to consult reference 26 as an example for a thoughtful evaluation of thrombosis and hemorrhage controlled by KLF2 and KLF4.

We thank the reviewer for this helpful comment. This very detailed in vivo study conducted by Sangwung et al focuses on the analysis of the spontaneous phenotype of the double KLF2/KLF4 KO adult mice and shows the redundant function of these two TF in vivo. In this JCI insight paper, the authors checked multiple parameters in vivo for evaluation of thrombosis and coagulopathy (D-dimer plasma levels, presence of schistocytes on blood smear, platelet and RBC counts, pTT/aPTT assays).

About the evaluation of clotting and bleeding, we have taken on board the Reviewer's comments and increased the evidence of thrombosis and coagulopathy in Erg^{IEC-KO} mice, as discussed above (increased number of mice analysed, fibrinogen/TAT ELISA, CD41 staining). We have carried out a similar characterization of the thrombotic phenotype; however we selected to use the CAT assay, as

this is reportedly a more sensitive assay compared to pTT and aPTT to measures thrombin generation in mouse plasma¹⁰.

*About the relationship between vascular disruption, hemorrhages and clotting: previous studies have shown that the loss of ERG is indeed associated with a level of blood vessels disruption (different architecture, increased vascular permeability¹¹; however, the integrity of the vasculature in these mice is not lost and not dramatically compromised. As pointed out by the Reviewer, extravascular blood can lead to clot formation; however, some *Erg*^{IEC-KO} mice (4/9) only present with clots and not hemorrhage in the liver. In line with this, the CAT assay on mouse plasma shows that loss of ERG in vivo leads to an increase in thrombin generation, suggesting a dysregulation of coagulation factors, leading to a prothrombotic state in these mice independent of endothelial integrity. We also hope that all the new in vivo data added to the manuscript provide convincing evidence of the prothrombotic phenotype in the *Erg*^{IEC-KO} mice.*

A general and strong concern regarding overall data interpretation (especially in vivo observations) is -as correctly pointed out by authors in the discussion p. 25- that conditional ablation of ERG in EC has pleiotropic effects, including lung and liver, that may largely be responsible for the observed phenotype. To establish a cause-effect relation between ERG-KLF2 interaction and thrombosis, it might be necessary to genetically remove the ERG-binding region in the proximal TM promoter that was detected in the CHIP assays and measure TM expression and thrombosis/fibrin deposition in different organs.

We appreciate the Reviewer's suggestion to perform a genetic removal of the ERG-binding region in the proximal TM promoter in vivo; this would be an elegant way to validate some of the molecular pathways identified here. However, this experiment requires the generation, validation and characterization of a new mouse line, which would inevitably delay this study for over 1 year. We considered carrying out a similar experiment on Zebrafish to reduce the time, but this is not possible since the TM proximal promoter containing the ERG binding sites is not conserved in Zebrafish.

*We believe that the new in vitro data firmly establishes the ERG-KLF2 relationship and its role in regulating TM, and that the experiments are in line with the state-of-the-art. We therefore focussed on demonstrating that TM was partly responsible for the coagulopathy observed in ERG-deficient mice and conducted an in vivo rescue experiment. We selected a recombinant red blood cells-targeted TM (RBC-TM), previously described by our new collaborators Prof Vladimir Muzykantov and Dr Raisa Kiseleva¹². This fusion protein was shown to have a more potent anticoagulant effect, with higher stability and half-life compared to soluble TM. RBC-TM was injected in control and *Erg*^{IEC-KO} mice, to see whether RBC-TM is able to partly rescue or normalize early plasma biomarkers of clotting/coagulopathy such as TAT.*

*We decided to carry out the rescue experiment in adult mice, 25 days post tamoxifen injection, since we have evidence of coagulopathy (increased TAT and D-dimer plasma levels) at this time point. Since the literature mainly reports short treatment protocols with recombinant TM in mouse models^{13,14}, we decided to perform an acute treatment by injecting a single dose of RBC-TM (IV, 4 mg/kg). Mice were bled before RBC-TM treatment and 6 hours after the injection to measure hemostasis-related plasma biomarkers. RBC-TM was able to rescue thrombin-antithrombin (TAT) levels in *Erg*^{IEC-KO} mice (Fig. 3b); this is an early marker of coagulopathy, with a short half-life (45 min), thus more likely to be sensitive to the acute intervention. We also measured other markers of coagulopathy, namely D-dimer (Fig. 3c) and fibrinogen (Fig. 3d); here, the acute RBC-TM treatment was not able to restore levels in *Erg*^{IEC-KO} mice. This is not unexpected, given the long half-lives of*

these proteins. These data show that an acute treatment with RBC-TM restores levels of early systemic plasma markers for coagulopathy/thrombosis (TAT). Controls for this experiment are presented in Fig. 3a and Supplementary Fig. 3a-e.

These new data demonstrate that loss of TM expression in Erg^{iEC-KO} mice is responsible for at least part of the prothrombotic phenotype.

Minor issues are:

- Page 6/results: The reference cited in results for the iEC-model (reference 36) used PDGFB-eGFP-Cre, while the methods state that Cdh5(PAC)-CreERT2 was used. Please clarify which one was used, and whether PDGFb- and Cad5 Cre have different phenotypes?

We apologize for the confusion, the reference (previously 36) for the Cdh5(PAC)-CreERT2 mice model was indeed wrong. The correct references are the following ones: Birdsey et al., Dev Cell, 2015¹⁵; Shah et al, Nat Comms, 2017¹¹. All the in vivo experiment conducted in the present study used the Cdh5(PAC)-CreERT2 driver line. The PDGFb-¹⁶ and Cdh5(PAC)- (unpublished data) Cre lines have a similar fibrotic phenotype. In this study, we chose to focus on the Cdh5(PAC)-Cre line phenotype and did not investigate the thrombotic phenotype in the PDGFb-Cre line. We made this choice in order to work with a Cre line without a GFP reporter that would reduce the possibilities of the fluorochromes we can use for immunostaining or for other types of in vivo experiments.

- Figure 4 a: the resolution of this image is insufficient to provide meaningful information (such as the sequence data provided in supplemental data).

We apologize for the low quality of the image. We have now improved the resolution of the TM promoter region presented in Fig.4 and added ERG ChIP-sequencing data on this region (recently published by our group⁷). The whole TM locus is also shown in Supplementary Fig. 4a.

Reviewer #3 (Remarks to the Author): *We thank the Reviewer for his/her comments*

This manuscript, Peghaire et al have identified an additional role for the endothelial ETS transcription factor ERG in vascular specific regulation of thrombosis. Along with its role in early endothelial cell development and angiogenesis, ERG has also been implicated in the maintenance of vascular integrity postnatally. Here, the authors describe how ERG, through its regulation of the anti-thrombotic receptor, Thrombomodulin, may prevent thrombosis during low shear stress conditions in vitro and in vivo.

In general, the data presented in the manuscript are detailed and rigorous and mostly support the overall conclusion of the study. The direct regulation of the TM promoter by ERG needs to be bolstered, and the direct relationship between ERG and KLF in the regulation of the TM promoter is not convincing. The authors should address my concerns, indicated below, prior to publication of this work.

Main concerns:

1) The authors conclude that Erg directly regulates TM via the ETS binding sites in the TM promoter. To demonstrate this more definitively, they should mutate the ETS binding sites in the TM promoter to determine if mutation of those sites results in a loss of Erg-dependent transactivation. As it is, the CHIP data provide circumstantial evidence for a direct, functional interaction, but otherwise the results could be interpreted as an indirect regulation of TM by Erg and there is no evidence presented that Erg binding is functional.

To prove that the ERG binding sites (EBS) in the TM promoter are functional, we followed the Reviewer's comment and performed mutagenesis of the EBS present in the TM proximal promoter. We generated two mutant TM promoter constructs (Supplementary Fig. 5b-c, Supplementary Fig. 6):

-mutant 1 with mutation of the 2 closest EBS, just upstream the TSS

-mutant 2 with mutation of all the EBS present on the TM promoter luciferase construct: 4 EBS in total

We conducted additional transactivation assays on HUVEC overexpressing ERG (or the pcDNA empty vector) using these 2 mutants and the wild type (WT), non-mutated TM promoter-luciferase construct. The data presented in Fig. 5c showed that ERG overexpression was able to transactivate the native TM promoter. Mutation of either the 2 EBS upstream the TSS or all the 4 EBS on the TM promoter resulted in loss of ERG-dependent transactivation (Fig. 5c), showing that the 2 EBS upstream the TSS are functional. These new data definitively show that ERG binding to the TM promoter is functional and that ERG directly drives TM expression.

2) Likewise, the conclusion that Erg recruits KLF to the TM promoter is not well supported. The proximity ligation assay demonstrates that the two factors are closely situated on chromatin but not necessarily in contact, and definitely do not support the idea that Erg recruits KLF to the promoter.

The data are more simply explained by the increase in H3K27Ac that is mediated by Erg, as shown by the authors' own data. Indeed, as the authors show, in the presence of high shear stress, Erg is not required for KLF binding, presumably because of increase H3K27Ac at TM caused by high shear. Therefore, to strengthen their results regarding the cooperative action of ERG and KLF in regulating the TM promoter, the authors should perform an additional transactivation assay to test whether the combined expression of ERG and KLF2 results in cooperative activation of the TM promoter cooperatively. Additionally, to supplement their CHIP-PCR experiment, the authors can also use the mutant TM promoter (mERG) from the previous point to determine if KLF transactivates the TM promoter in the absence of Erg binding sites.

To address the Reviewer's concern, we carried out the suggested experiments on the ERG-KLF2 cooperation in driving TM expression. We performed additional assays on HUVEC overexpressing ERG or KLF2 and treated with KLF2 or ERG siRNA. qPCR (Supplementary Fig. 8f) and transactivation assay (Fig. 6a) revealed that KLF2 promotes ERG's ability to drive the TM promoter and its expression. On the other hand, KLF2 overexpression (Supplementary Fig. 8g and h) was able to drive the TM promoter (Fig. 6b) and its expression (Supplementary Fig. 8g). Loss of ERG expression abrogated this effect, showing that ERG is required for KLF2 to transactivate the TM promoter (Fig. 6b and Supplementary Fig. 8g). Moreover, overexpression of ERG and KLF2 was able to further transactivate the TM promoter compared to single ERG or KLF2 overexpression condition (Fig. 6c), confirming cooperativity of these two TF.

We also followed the Reviewer's comment and performed new transactivation assays using the mutant TM promoter constructs (mutated for ETS binding sites -EBS) discussed in the previous point, on HUVEC overexpressing KLF2 (or the pcDNA empty vector). The data presented in Fig. 6d shows that KLF2 overexpression was able to transactivate the TM WT promoter; however, mutation of either the 2 EBS upstream the TSS or the 4 EBS in the TM promoter completely abrogated the ability of KLF2 to transactivate the TM promoter. These new data prove that KLF2 requires ERG and the EBS upstream of the TSS to bind to the TM promoter and to transactivate TM expression, confirming ERG and KLF2 direct cooperation in driving TM expression.

4) In the presence of high shear forces, in the presence and absence of Erg, does H3K27Ac at the TM promoter increase? This needs to be tested to provide some mechanistic insight into the requirement of Erg for TM expression and subsequent regulation of thrombosis.

We thank the Reviewer for this comment and we have added new data to address the role of ERG in the regulation of H3K27Ac. The acetyltransferase p300 has been shown to bind to TM promoter and promotes its expression⁶. Moreover, our group has recently shown that ERG is involved in p300 recruitment and acetylation of H3K27 on super-enhancers associated with endothelial cells-specific genes⁷. Our new data show that pharmacological inhibition of p300 leads to a decrease in H3K27Ac occupancy on the TM promoter (Fig. 5d and Supplementary Fig.7a) and to a decrease in TM mRNA levels (Supplementary Fig. 7b). Moreover, our new ChIP-qPCR data presented in Fig. 5e shows that deletion of ERG in HUVEC leads to a significant decrease in p300 occupancy associated with decreased H3K27Ac levels (Fig. 5f) on TM promoter. These data prove that ERG is required for p300 recruitment, leading to the acetylation of H3K27 and opening the chromatin on the TM promoter region to allow transcription.

We found the Reviewer's suggestion to look at H3K27Ac levels at the TM promoter (in the presence or absence of ERG) in high shear stress (SS) conditions very interesting, but technically challenging. One way to address this particular question would be to perform H3K27Ac ChIP-qPCR in HUVEC treated with control or ERG siRNA and cultured under flow. Unfortunately, reliable ChIP-qPCR protocols require 1-2 million cells and our flow system (Ibidi) uses micro-slides that can only contain 150000 endothelial cells (maximum). Therefore, this experiment in primary endothelial cells is technically challenging.

As an alternative experiment to address this point, we decided to look at H3K27Ac global levels by performing immunofluorescence microscopy for H3K27Ac on HUVEC treated with control or ERG siRNA and exposed to static, low or high SS conditions for 24 hours (new data presented in Fig. 7e). In line with our recent data⁷, H3K27Ac global levels were decreased in ERG-deficient HUVEC compared to control siRNA (Fig. 7e). Low SS and high SS increased acetylation of H3K27Ac in control HUVEC compared to static conditions (Fig. 7e), as expected. Notably, levels of H3K27Ac in LSS conditions were significantly decreased in ERG-deficient cells compared to HUVEC treated with control siRNA, whereas H3K27Ac levels under HSS conditions were not affected by the loss of ERG (Fig. 7e).

These data suggest that ERG contributes to the chromatin accessibility in static and low SS conditions, i.e. in conditions where ERG is required for the regulation of TM expression, but is not a key regulator of this process under high SS conditions where ERG is dispensable for TM expression. We speculate that transcription factors other than ERG (or other cofactors or epigenetic mechanisms) promote chromatin accessibility and TM expression in high SS conditions^{17,18,19,20}.

5) (Minor) In Figure 4A, it would be useful to show the entire Thrombomodulin locus, as well as the zoomed view of the region of interest.

We are happy to comply with the Reviewer's request. We have improved the resolution of the TM promoter region presented in Fig. 5a and added ERG ChIP-sequencing data⁷ on this region. We also included the whole TM locus in Supplementary Fig. 5a.

References

1. Sheikh S, Ed Rainger G, Gale Z, Rahman M, Nash GB. Exposure to fluid shear stress modulates the ability of endothelial cells to recruit neutrophils in response to tumor necrosis factor- α : a basis for local variations in vascular sensitivity to inflammation. *Blood*. 2003;102:2828-2834. doi:10.1182/blood-2003-01-0080
2. HH. L. Shear stress in the circulation. In: *Bevan J, Kaley G, Eds. Flow Dependent Regulation of Vascular Function*. New York: Oxford University Press; 1985:28-45.
3. Aird WC. Phenotypic heterogeneity of the endothelium: II. Representative vascular beds. *Circ Res*. 2007;100(2):174-190. doi:10.1161/01.RES.0000255690.03436.ae
4. Reneman RS, Hoeks APG. Wall shear stress as measured in vivo: Consequences for the design of the arterial system. *Med Biol Eng Comput*. 2008;46(5):499-507. doi:10.1007/s11517-008-0330-2
5. Suo J, Ferrara DE, Sorescu D, Guldberg RE, Taylor WR, Giddens DP. Hemodynamic shear stresses in mouse aortas: Implications for atherogenesis. *Arterioscler Thromb Vasc Biol*. 2007;27(2):346-351. doi:10.1161/01.ATV.0000253492.45717.46
6. Sohn RH, Deming CB, Johns DC, et al. Regulation of endothelial thrombomodulin expression by inflammatory cytokines is mediated by activation of nuclear factor-kappa B. *Blood*. 2005;105(10):3910-3917. doi:10.1182/blood-2004-03-0928
7. Kalna V, Yang Y, Peghaire C, et al. The Transcription Factor ERG Regulates Super-Enhancers Associated with an Endothelial-Specific Gene Expression Program. *Circ Res*. 2019;124 (9):1337-1349. doi:10.1161/CIRCRESAHA.118.313788
8. Stalker TJ, Traxler EA, Wu J, et al. Hierarchical organization in the hemostatic response and its relationship to the platelet-signaling network. *Blood*. 2013;121(10):1875-1885. doi:10.1182/blood-2012-09-457739
9. Isermann B, Hendrickson SB, Zogg M, et al. Endothelium-specific loss of murine thrombomodulin disrupts the protein C anticoagulant pathway and causes juvenile-onset thrombosis. *J Clin Invest*. 2001;108(4):537-546. doi:10.1172/JCI200113077.Introduction
10. Kasthuri RS, Glover SL, Boles J, Mackman N. Tissue Factor and Tissue Factor Pathway Inhibitor as Key Regulators of Global Hemostasis: Measurement of Their Levels in Coagulation Assays. *Semin Thromb Hemost*. 2011;36(7):764-771. doi:10.1055/s-0030-1265293.Tissue
11. Shah A V., Birdsey GM, Peghaire C, et al. The endothelial transcription factor ERG mediates Angiopoietin-1-dependent control of Notch signalling and vascular stability. *Nat Commun*. 2017;8(May):1-16. doi:10.1038/ncomms16002
12. Zaitsev S, Kowalska MA, Neyman M, et al. Targeting recombinant thrombomodulin fusion protein to red blood cells provides multifaceted thromboprophylaxis. *Blood*. 2012;119(20):4779-4785. doi:10.1182/blood-2011-12-398149
13. Suyama K, Kawasaki Y, Miyazaki K, et al. The efficacy of recombinant human soluble thrombomodulin for the treatment of shiga toxin-associated hemolytic uremic syndrome model mice. *Nephrol Dial Transplant*. 2015;30(6):969-977. doi:10.1093/ndt/gfv004
14. Ding B Sen, Hong N, Christofidou-Solomidou M, et al. Anchoring fusion thrombomodulin to the endothelial lumen protects against injury-induced lung thrombosis and inflammation. *Am J Respir Crit Care Med*. 2009;180(3):247-256. doi:10.1164/rccm.200809-1433OC
15. Birdsey GM, Shah A V, Dufton N, et al. The endothelial transcription factor erg promotes vascular stability and growth through Wnt/ β -catenin signaling. *Dev Cell*. 2015;32(1):82-96. doi:10.1016/j.devcel.2014.11.016
16. Dufton NP, Peghaire CR, Osuna-Almagro L, et al. Dynamic regulation of canonical TGF β signalling by endothelial transcription factor ERG protects from liver fibrogenesis. *Nat Commun*. 2017;8(1):1-14.

- doi:10.1038/s41467-017-01169-0
17. Jiang YZ, Manduchi E, Jiménez JM, Davies PF. Endothelial Epigenetics in Biomechanical Stress: Disturbed Flow-Mediated Epigenomic Plasticity in Vivo and in Vitro. *Arterioscler Thromb Vasc Biol.* 2015;35(6):1317-1326. doi:10.1161/ATVBAHA.115.303427
 18. Illi B, Nanni S, Scopece A, et al. Shear Stress-Mediated Chromatin Remodeling Provides Molecular Basis for Flow-Dependent Regulation of Gene Expression. *Circ Res.* 2003;93:155-161. <http://www.circresaha.org>. Accessed April 29, 2019.
 19. Von Der Ahe D, Nischan C, Kunz C, et al. *Ets Transcription Factor Binding Site Is Required for Positive and TNFa-Induced Negative Promoter Regulation.* Vol 21.; 1993.
 20. Milkiewicz M, Uchida C, Gee E, Fudalewski T, Haas TL. Shear stress-induced Ets-1 modulates protease inhibitor expression in microvascular endothelial cells. *J Cell Physiol.* 2008;217(2):502-510. doi:10.1002/jcp.21526

REVIEWERS' COMMENTS:

Reviewer #1 (Remarks to the Author):

The authors have addressed my concerns. However, my opinion that the manuscript is incrementally novel remains.

Reviewer #2 (Remarks to the Author):

The authors have responded in an exhaustive and constructive manner to my comments. The substantial additional data, together with modifications in text and figures are adequate for clarification and further support of the proposed mechanism. Taken together, the body of additional work for addressing my own and other reviewer's comments seem adequate to assure rigor of the study. While it doesn't provide paradigm-changing conceptual insights, it nevertheless is a very thorough extension of previous data and the potential in vivo relevance of these data.

Reviewer #3 (Remarks to the Author):

My concerns have been adequately addressed.

Reply to reviewers: Manuscript NCOMMS-18-24141B

The transcription factor ERG regulates a low shear stress-induced anti-thrombotic pathway in the microvasculature

Reviewers' comments:

Reviewer #1 (Remarks to the Author):

The authors have addressed my concerns.

We thank Reviewer #1 for her/his comment.

However, my opinion that the manuscript is incrementally novel remains.

We still disagree with the view that our study is not showing enough novelty.

Reviewer #2 (Remarks to the Author):

The authors have responded in an exhaustive and constructive manner to my comments. The substantial additional data, together with modifications in text and figures are adequate for clarification and further support of the proposed mechanism. Taken together, the body of additional work for addressing my own and other reviewer's comments seem adequate to assure rigor of the study. While it doesn't provide paradigm-changing conceptual insights, it nevertheless is a very thorough extension of previous data and the potential in vivo relevance of these data.

We thank Reviewer #2 for her/his very kind words and for considering the manuscript acceptable for publication in Nature Communications.

Reviewer #3 (Remarks to the Author):

My concerns have been adequately addressed.

We thank Reviewer #3 for considering the manuscript acceptable for publication in Nature Communications.